# Natural epialleles of *Arabidopsis SUPERMAN* display *superwoman* phenotypes

Ramesh Bondada 📷 1, Saravanakumar Somasundaram 📷 1, Mohan Premanand Marimuthu2,
Mohammed Afsal Badarudeen1, Vaishak Kanjirakol Puthiyaveedu1 & Ravi Maruthachalam 📷 1✉

Epimutations are heritable changes in gene function due to loss or gain of DNA cytosine methylation or chromatin modifications without changes in the DNA sequence. Only a few natural epimutations displaying discernible phenotypes are documented in plants. Here, we report natural epimutations in the cadastral gene, *SUPERMAN*(*SUP*), showing striking phenotypes despite normal transcription, discovered in a natural tetraploid, and subsequently in eleven diploid *Arabidopsis* genetic accessions. This natural *lois lane*(*lol*) epialleles behave as recessive mendelian alleles displaying a spectrum of silent to strong *superwoman* phenotypes affecting only the carpel whorl, in contrast to semi-dominant *superman* or *supersex* features manifested by induced epialleles which affect both stamen and carpel whorls. Despite its unknown origin, natural *lol* epialleles are subjected to the same epigenetic regulation as induced *clk* epialleles. The existence of *superwoman* epialleles in diverse wild populations is interpreted in the light of the evolution of unisexuality in plants.

1 School of Biology, Indian Institute of Science Education and Research (IISER)-Thiruvananthapuram, Vithura, Kerala 695551, India. 2 Genome Center, University of California, Davis, CA 95616, USA. ✉email: ravi@iisertvm.ac.in

The *Arabidopsis thaliana* hermaphrodite (bisexual) flower consists of four concentric whorls of floral organs: outer (first) whorl composed of four sepals, second whorl with four petals, third, male reproductive whorl with six stamens and the innermost (fourth), female reproductive whorl consisting of two fused carpels forming a single bicarpellary pistil. A cohort of genes with distinct spatiotemporal expression in the floral meristem (FM) ensures an orderly establishment of the floral whorls[1]. The *SUPERMAN* is one such gene that encodes a C2H2 zinc finger domain containing transcription factor regulating floral homeotic genes during *Arabidopsis* flower development[2–4]. The *SUP* is essential for demarcating the floral boundary between the stamen and carpel whorls, FM termination, carpel-placenta boundary specification, and ovule integument differentiation[3–8]. SUPERMAN function is regulated both at genetic and epigenetic levels, as demonstrated by the identification and characterization of several genetic mutants and induced epimutants (*clark kent* (*clk*) epialleles) with varying phenotypic consequences. Based on the phenotypic variations in sexual boundary distortions in the reproductive whorls and FM indeterminacy, *sup* (epi)mutants have been classified as (i) *superman* (whorl 3 indeterminacy leading to male flowers with supernumerary stamens at the expense of carpels), (ii) *superwoman* (whorl 4 indeterminacy producing more than two carpels, with unaffected stamen whorl), and (iii) *supersex* (both whorl 3 and whorl 4 indeterminacy producing supernumerary stamens and carpels)[6].

Epialleles arise due to methylation of cytosines in a gene loci without any change in the DNA sequence[9]. In animals, the cytosine methylation is mostly confined to symmetric CG context, with some exceptions[10,11]. Whereas in plants, it occurs in all sequence contexts: CG, CHG (H = A, T, C), and CHH[12]. Epimutations, natural or induced, add a new layer of heritable variation in addition to natural genetic polymorphisms existing in the germplasm, influencing many plant traits[13–20] and may have a role in adaptive evolution[21]. However, only a limited number of natural epimutations with phenotypic consequences are reported in the plant kingdom[22,23]. The resultant phenotypes in those natural epimutants are attributed to cytosine methylation of the regulatory (promoter) regions that negatively affect the transcription of the respective gene loci, resulting in either a total loss or reduced mRNA levels[24]. The *clk* epialleles of *SUP* are the first characterized induced epimutations in *A. thaliana*, whose genetic locus is shown to harbor methylated cytosines in all three sequence contexts[25]. Unlike natural epimutants, epigenetic regulation of *SUPERMAN* is atypical in that the methylated cytosines are seen predominantly in the transcribed regions of the loci rather than to its regulatory regions[25]. In the well-characterized induced *sup* epiallele *clk-3*, the *superman* phenotype is attributed to a reduction in the levels of mRNA[25], whereas in some induced *sup* epialleles, no reductions in mRNA are observed despite DNA hypermethylation[6] suggesting complex epigenetic regulation of the *SUP* gene.

In plants, the de novo methylation of genomic loci is established by siRNA mediated RNA dependent DNA methylation pathway (RdDM)[26], whose molecular machinery consists of a multitude of proteins such as plant-specific RNA polymerases Pol IV and Pol V, *DICER-LIKE 2 (DCL2), ARGONAUTE4 (AGO4), DOMAINS REARRANGED METHYLTRANSFERASE 2 (DRM2)*[12,27]. DRM2 recruited to the corresponding loci by the RdDM machinery methylates the cytosines. Once established, methylated cytosines at CG context is maintained by DNA METHYLTRANSFERASE (*MET1* aka *DNMT* in humans)[28,29], while *DRM2*, CHROMO-METHYLASE 2 (*CMT2*), and CHROMOMETHYLASE 3 (*CMT3*) control methylation at CHG and CHH contexts in a partially redundant and locus-specific manner[30–33]. The DNA methyltransferases CMT2 and CMT3 are recruited to regions enriched for H3K9Me, which is methylated by a histone methyltransferase *KRYPTONITE (KYP)* in a self-reinforcing feedback loop linking both DNA and histone methylation[34]. In fact, *SUP* epimutants played a crucial role in discovering *CMT3*, *KYP*, and *AGO4* genes as being extragenic suppressors of *clk* epimutations[31,35,36]. Unlike *CMT3* and *KYP*, mutations in methyltransferases *DRM1*, *DRM2*, *MET1*, and the chromatin remodeler DECREASE IN DNA METHYLATION 1 (*DDM1*) are not required for maintaining the methylation at *SUP* locus[37,38]. The prevailing model for *SUP* locus methylation deciphered based on the study of induced (*clk*) epialleles suggest that AGO4 guides KYP to *SUP* chromatin to methylate H3K9, which creates a binding platform for LIKE HETEROCHROMATIN PROTEIN 1 (LHP1), which in turn recruits CMT3 catalyzing the methylation of cytosines at *SUP* locus, converting the *SUP* WT allele to *clk* epiallele[36]. Thus, the methylation of *SUP* locus in epimutants is independent of the RdDM pathway where the recruitment of the DRM2 mediates DNA methylation.

Intriguingly, *SUP* locus is unusual in the preponderance of induced epialleles ($n = 12$) in comparison to genetic alleles ($n = 6$) reported for any gene loci in *A. thaliana* (Supplementary Table 1). Despite these reports, no instance of natural epialleles has been discovered to date. Here, we report the existence of natural *SUP* epialleles in 12 of the 1028 core germplasm accessions (representing the global genetic diversity in *A. thaliana*) examined, mostly displaying *superwoman* phenotypes in contrast to *superman* phenotypes observed in induced epialleles, further deepening the mystery behind epigenetic regulation of *SUP*. We interpret the predominance of natural *superwoman* epialleles of *SUPERMAN* in the wild populations in the context of the evolution of unisexuality in flowering plants.

## Results

**Phenotypic characterization of *superwoman* inflorescences in the natural tetraploid Wa-1 *Arabidopsis* accession.** The species *A.thaliana* prevails in the wild as diploid; however, rare tetraploid populations do exist[39]. One such tetraploid accession($4x = 2n = 20$, $x$ = basic chromosome number 5), Warschau-1 (Wa-1, CS22644) caught our attention because of its striking heterogeneity in the inflorescence phenotypes displaying a mix of normal bilocular siliques, unusual multilocular siliques, and curly siliques (Fig. 1a–g). The phenotype is 100% penetrant ($n = 1080$ plants) but with variable expressivity between individuals. We broadly classified the plants ($n = 1080$) into three categories based on the inflorescence phenotype: (i) plants with inflorescences predominantly displaying multilocular siliques (56%; Fig. 1a), (ii) plants with more curly siliques (9.5%; Fig. 1b), and (iii) plants with a random distribution of all types in varying proportions (34.2%; Fig. 1c). Overall, up to 63% ($n = 3402/5400$) of the siliques are multilocular than normal ($n = 864/5400, 16\%$), with remaining being curly ($n = 1134/5400, 21\%$). To check whether these inflorescence phenotypes are transmitted true to type, we harvested seeds from each category and scored for the transmissibility of the respective phenotypes in the immediate progeny. Irrespective of the category, we observed all the three types of inflorescence reappearing in random proportions, suggestive of variable expressivity, however, with 100% penetrance (Supplementary Table 2).

A closer inspection of the multilocular siliques revealed that they had more than two fused carpels. The phenotype was more apparent in the transverse sections (Fig. 1e–g), matured siliques (Fig. 1h, i), and in the replum–septum skeleton of dehisced siliques (Fig. 1j). A typical WT silique consists of two replums, a contiguous membranous septum joining two replums, two placental arrays of seeds originating on the inner sides of either

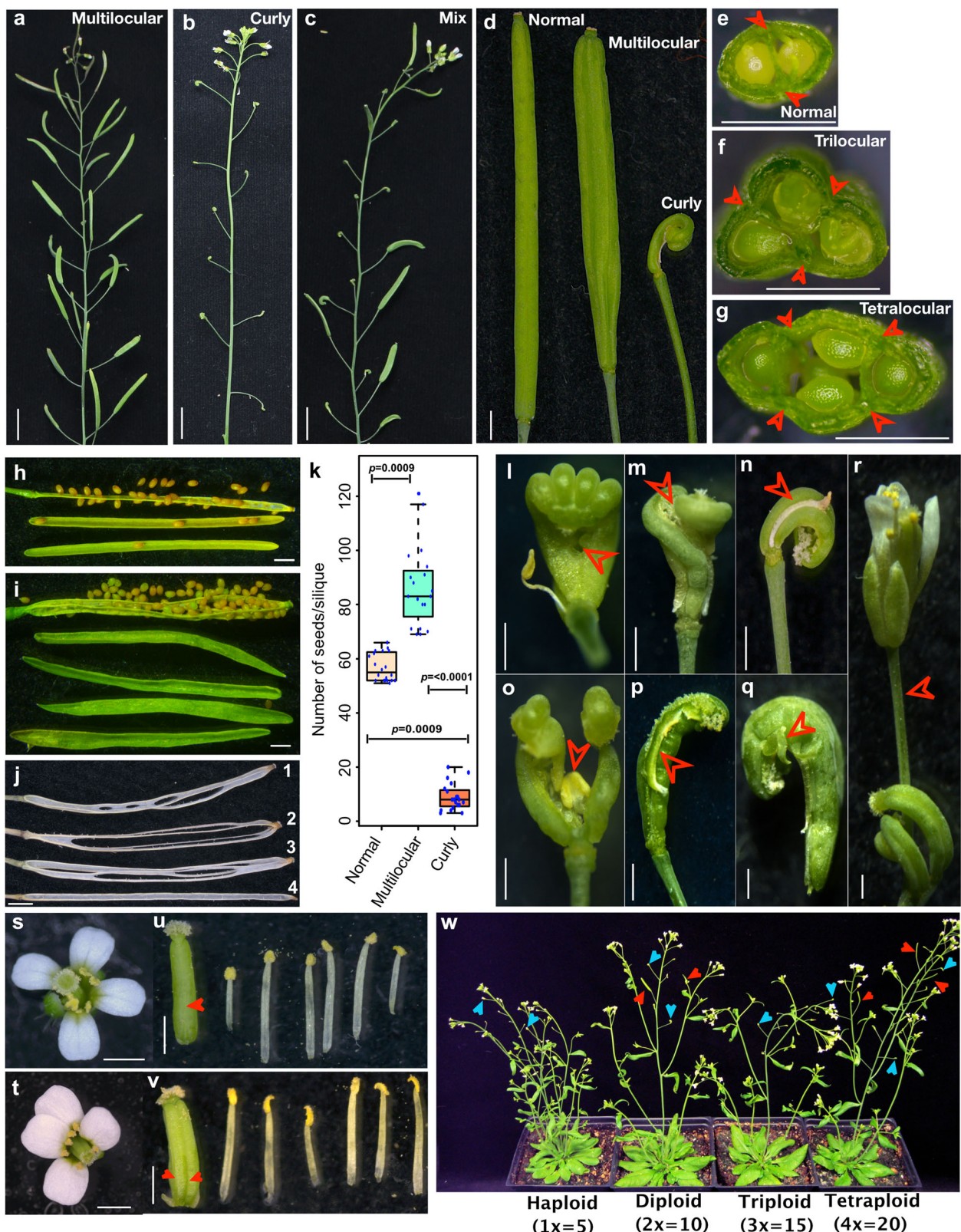

replum, and two valves protecting the seeds (Fig. 1h and Supplementary Fig. 1). In contrast, the multilocular siliques contained more than two replums, two or more discontinuous septum's, three or more placental arrays of seeds, and multiple carpel valves (Fig. 1i, j and Supplementary Fig. 1). Despite these silique patterning defects, the seed set in the multilocular siliques was not only normal but super fertile (hence *superwoman*) with

69–121 seeds/silique ($n = 20$) in contrast to its normal (WT) counterpart containing 52–64 seeds/silique ($n = 20$) (Fig. 1k). The curly siliques are due to abnormal fusion of supernumerary carpels (Fig. 1l, m) and are either sterile or partially fertile with 2–21 seeds/silique ($n = 20$). In some pistils, we observed organ fusion defects (Fig. 1n), and various homeotic transformations, as shown in Fig. 1o–r and they remained sterile without any seeds.

**Fig. 1 Phenotypic characterization of inflorescence from tetraploid Wa-1 plants. a–c** Representative images of inflorescence types observed in the tetraploid (4x) Wa-1 population. **a** An inflorescence with super fertile multilocular siliques. **b** An inflorescence consisting only of curly pistils/siliques. **c** An inflorescence containing a random distribution of both multilocular and curly siliques. **d** Representative images of a Wa-1 normal silique arising from a bicarpellary WT pistil, a multilocular silique originating from a pistil with supernumerary carpel and a developmentally arrested curly pistil/silique. **e–g** A transverse section of a bicarpellary (**e**), tricarpellary (**f**), and a tetracarpellary (**g**) silique from Wa-1 plants. Arrowheads indicate replum. **h, i** Matured siliques from normal (**h**) and multilocular (**i**) Wa-1 split open to reveal its anatomy. Normal silique shows two placental arrays of seeds attached to either replum surrounded by two carpel valves (**h**). Multilocular silique shows four placental arrays of seeds on a multiple repla surrounded by four carpel valves (**i**). **j** Replum-septum skeletons of individual siliques post dehiscence depicting multiple irregular replums and abnormal fusion of septum in multilocular siliques (numbered 1, 2, 3) to the normal one having two parallel replums interconnected by a continuous membranous septum (numbered 4). **k** Box and whisker plot showing seed count/silique from random ($n = 20$) normal, multilocular and curly siliques, respectively. The seed set is statistically significant ($p < 0.0001$, Kruskal–Wallis test, 2df, $\alpha = 0.05$, Dunn's posthoc test for three pairwise comparisons are also statistically significant with $p$ values as indicated. **l** A partially fused (arrowhead) tetracarpellary pistil posing a folded fingers like phenotype. **m** Unfused tetracarpellary pistil showcasing two partially fused bicarpellary pistils embracing each other. **n** Congenital fusion of stamen to improperly fused carpel. **o** A stamen (arrowhead) arising from the carpel tissues. **p** Proliferative mass of apical stigmatic like tissues seen on the median portion of the carpel. **q** Sepal-like leafy outgrowth in carpels. **r** A flower originating from carpel tissues. **s** WT flower from the diploid Col-0 accession. **t** A *superwoman* flower from the tetraploid Wa-1 accession. **u** WT flower shown in **s** dissected to reveal six stamens and a bicarpellary pistil(arrowhead). **v** The *superwoman* flower shown in **t** dissected to reveal six stamens and a tetracarpellary pistil (arrowhead). **w** Wa-1 ploidy series consisting of haploid (x), diploid (2x), triploid (3x), and tetraploid (4x) individuals showing Wa-*superwoman* phenotypes (red arrowhead—multilocular siliques, blue arrows—curly siliques). Scale bar: **a–c** = 1 cm, **d–t** = 1 mm.

Taken together, the phenotypes observed in Wa-1 (hereafter Wa-*superwoman*) are indicative of a floral homeotic transformation, explicitly affecting the fourth(carpel) whorl, with no visible phenotypes on the remaining three whorls of the flower (Fig. 1s–v). We also confirmed similar phenotypes in several other Wa-1 germplasm stocks CS28805, CS39006, CS6885, and CS1586 available at Arabidopsis Biological Resource Centre (ABRC). Hence, we conclude that the Wa-*superwoman* phenotype is characteristic of the Wa-1 genotype with an underlying genetic cause affecting floral organ patterning.

**Genetic mapping and complementation identifies *SUPERMAN* as the causative locus behind *Wa-superwoman* phenotype.** To unequivocally identify the locus, we employed a map-based cloning approach. However, the generation of a mapping population in the Wa-1 tetraploid is challenging due to the complex tetrasomic segregation of alleles. Hence, for mapping and subsequent genetic experiments, we used Wa-1 diploid (2x) derived from the Wa-1 tetraploid using *CenH3* based genome elimination system[40]. To this end, we confirmed the penetrance and reproducibility of the Wa-*superwoman* phenotypes in the derived diploid population of Wa-1 plants ($n = 950$, 100% penetrance, Supplementary Fig. 2). In addition, we also observed identical inflorescence phenotypes in all haploid (1x) plants ($n = 41$) and triploid (3x) Wa-1 plants ($n = 62$), implying that the phenotype is independent of gene or genome dosage (Fig. 1w).

To generate a mapping population, we first crossed the derived diploid Wa-1 to WT Col-0 and L*er* accessions. Irrespective of the accessions, all tested F1 progeny ($n = 120$) showed normal pistil and siliques that were indistinguishable from the WT, indicating the recessive nature of the locus (Supplementary Fig. 3a). Since Col-0 is more polymorphic to Wa-1 than L*er* accession, we advanced the Wa-1 × Col-0 F1's to F2 generation for subsequent mapping. In the F2 population, the phenotype segregated in a 3:1 (394 WT:126 Wa-*superwoman* $\chi^2 = 0.083$, 1 df, $p = 0.77$) Mendelian ratio confirming its monogenic recessive nature. Next, we genotyped the F2 segregants showing the Wa-*superwoman* phenotype using genome-wide SSR markers (Supplementary Fig. 3b) and identified a marker MSAT3.19 in the third chromosome tightly linked to the phenotype ($n = 90/100$ plants homozygous for Wa-1 polymorphism). Among the genes located in the mapped region, we chose the *SUP locus* (~566 kb from MSAT 3.19) as a promising candidate due to the Wa-*superwoman's* phenotypic resemblance with floral phenotypes exhibited by *superman* mutants[6,41]. At the DNA sequence level, except

for a single nucleotide polymorphism (G to A) at 300 bp upstream of the *SUP* start codon, *Wa-1 SUP* locus (6.7 kb complementing region) was otherwise indistinguishable from WT Columbia (Col-0) reference sequence (TAIR 10). Hence, we converted this single nucleotide polymorphism (SNP) as a dCAPS (derived Cleaved Amplified Polymorphic Sequences) marker[42], and genotyped the F2 segregants showing the Wa-*superwoman* phenotype and found the SNP to be co-segregating (100%, $n = 100$) with the phenotype, suggesting that *SUP* is the causative locus (Supplementary Fig. 3c).

To further validate that the identified gene locus is *SUP*, we performed two genetic tests: an allelism test by crossing a homozygous recessive *sup-5* deletion mutant (displaying *supersex* phenotype) with Wa-1 diploid and showed that all the F1 plants ($n = 93$) had the otherwise recessive *superwoman* phenotype (Supplementary Fig. 4a), proving their allelic nature. Similar is the case in the F1 progeny harboring different heteroallelic combinations: Wa-1 allele with *clk-3* epiallele ($n = 24$, Supplementary Fig. 4b) and Wa-1 allele with *sup-2* truncated allele ($n = 15$, Supplementary Fig. 4c). Secondly, the genetic complementation test: a 6.7 kb genomic clone containing the *SUP* locus inserted elsewhere in the *sup* genetic and epigenetic mutants is shown to complement the *sup* mutation[2,25]. Hence, we cloned the corresponding region from Wa-1 harboring the SNP and introduced it into a *sup-5* mutant and tetraploid Wa-1 plants, and found that it completely rescues the mutant to WT phenotype in all the T1 plants examined both in *sup-5* ($n = 8$) and Wa-1 tetraploid ($n = 7$, Supplementary Fig. 4d). We observed similar WT phenotypes when the corresponding Col-0 genomic *SUP* WT locus was introduced into the tetraploid Wa-1 plants ($n = 11$, Supplementary Fig. 4d). Altogether, this rules out DNA sequence change (SNP) in the *SUP* locus as the cause of the Wa-*superwoman* phenotype.

**DNA methylation analysis identifies *lois lane* (*lol*) natural epiallele of *SUPERMAN* in Wa-1.** Given, there are no genetic alterations in the Wa-1 *SUP* genomic region, and the locus is known to be hypermethylated in induced epialleles, we examined its methylation status in the diploid Wa-1 plants by bisulfite sequencing. We found that a stretch of the Wa-1 *SUP* transcribed region ($-282$ to $+776$) was hypermethylated (Fig. 2a, b and Supplementary Fig. 5), similar to that of induced *clk* epialleles of *SUP*[25,41]. This was further validated by Chop PCR, an assay integrating the use of methylation-sensitive restriction enzymes (MSRE) with PCR to detect the methylation state of any genomic

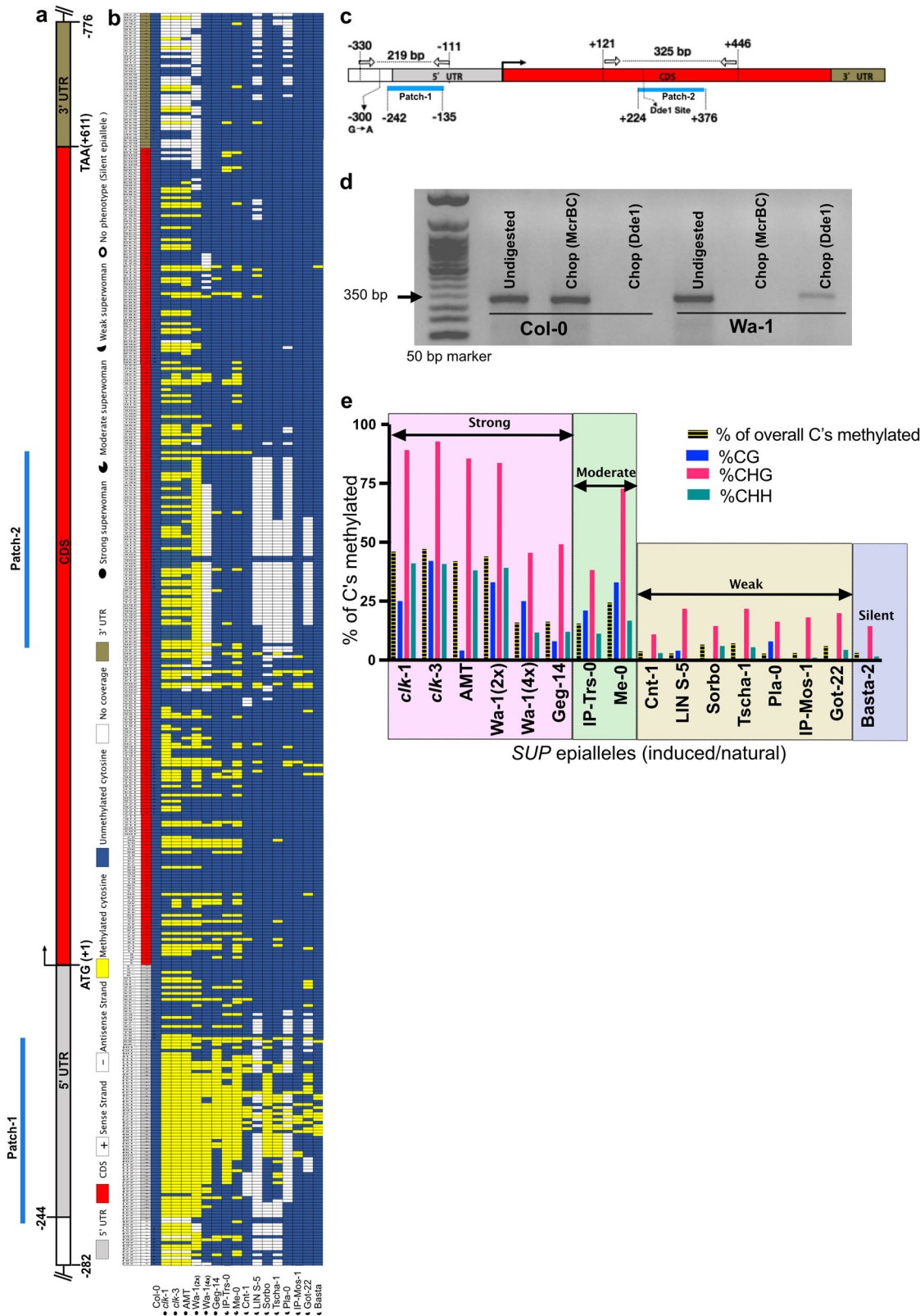

locus[19,43,44]. We chose a densely methylated region (patch-2) in *SUP* locus (Fig. 2a, c) in a way that it contains at least one cleavage site for two complementary MSREs: 1. McrBC (cleaves within the patch if any methylcytosine is preceded by a purine (A/ G) on one or both the strands when two such sites are separated by a distance of 40–3000 bp) and 2. Dde1 (protects the patch from cleavage when cytosine is methylated in its recognition sequence 5′CTNAG3′). As expected, McrBC completely cleaved the region resulting in no PCR amplification with the primers flanking patch-2 region, whereas Dde1 protected the same part from cleavage resulting in PCR amplification in comparison to unmethylated Col-0 control consistent with the presence of methylated cytosines at Wa-1 *SUP* locus (Fig. 2d and Supplementary Fig. 13a). The faint PCR signal observed in the Dde1

**Fig. 2 Methylation profiles of cytosines in the top (+) and the bottom (−) strands of *SUP* gene locus. a** A cartoon representation of the hypermethylated *SUP* transcribed region. The numbers depict the nucleotide sequence's relative position with reference to adenine nucleotide in ATG start codon of *SUP* gene as +1. **b** Methylation status of all cytosines present in the *SUP* transcribed region read from the bottom (5′ UTR) to top (3′ UTR). Individual cytosine methylation profiles for 17 genotypes are shown in the order as in **e**: Col-0 as unmethylated control, *clk-1*, *clk-3*, AMT (antisense methyltransferase1) induced *sup* epialleles as methylated controls (data for the figure is extracted from Fig. 3[25], 2x Wa-1(our high throughput sequencing data, two biological replicates), 12 natural accessions including Wa-1 (2*n* = 4*x*) are from the 1001 epigenomes project[21]. The cytosine methylation data for these 12 accessions are extracted from the 1001 epigenomes database. (+) for the sense strand and (−) for the antisense strand. Methylated and unmethylated cytosines are color-coded, as shown in the legend. White boxes indicate cytosines which didn't have enough coverage to call it methylated or not. The symbols at the beginning of each column indicate the *sup* natural epiallele's phenotypic strength displayed by the respective accession, as shown in the legend. An expanded view of figure panel **b**, is shown in Supplementary Fig. 5 for better clarity. **c** Cartoon representation of *SUP* locus depicting the patch-1 and patch-2 region and corresponding flanking primers(forward and reverse arrows) used for dCAPS assay and Chop PCR. **d** Chop PCR assay for patch-2 in unmethylated Col-0 control and Wa-1 (2*x*) using complementary MSRE: McrBC and Dde1. **e** A bar graph showing the % of cytosines methylated in different sequence contexts for the 17 genotypes represented in **b**. The individual accessions are shown on the *x*-axis, and the *y*-axis represents the % of methylated cytosines. Each genotype is represented by four vertical bars. The first bar represents the % of overall cytosines methylated at *SUP* transcribed region spanning −282 to +869 (1151 bp), consisting of 445 cytosines in total from both + and − strands. Overall there are 366 cytosines in CHH context, 55 in CHG context, and 24 in CG context in the transcribed region. The remaining three bars represent % of cytosines methylated, respectively, at CG, CHG, and CHH (where H = C, A, T) context. The phenotypic strength of natural epialleles (strong, moderate, and weak Wa-*superwoman* phenotypes) is roughly proportional to the % of cytosines methylated as indicated.

digest may be suggestive of heterogeneous methylation of that cytosine residue in the cells of the inflorescence tissue. Taken together, we have discovered a natural epiallele of *SUP*, which we named as *lois lane* (*lol-1*), after the fictional superwoman comic character (in coherence with the naming convention of artificially induced *clk* epialleles of *SUPERMAN* after Clark Kent character). Although the gene body of *SUP* locus is hypermethylated, its methylation pattern is distinct from the gene body methylated (gbM) loci. Unlike canonical gbM loci enriched for CG methylation[21], *SUP* epialleles are biased to high CHG and CHH methylation with negligible CG methylation in its gene body (Fig. 2e). Such methylation patterns are characteristic of transposon element methylation (teM) loci[21]. However, we didn't find any unique transposable elements specific to Wa-1 *SUP* locus in its vicinity, at least ~50 kb in either direction, as the DNA sequence is identical to the unmethylated Col-0 genome except for a few SNPs. A majority of the methylated cytosines are common to *clk* and *lol-1* epialleles (~80%, Supplementary Table 3) except for a few, especially in the patch-2 region and others being scattered throughout the locus (Fig. 2b and Supplementary Fig. 5). Other than this, we didn't find any characteristic methylation patterns unique to *lol-1* epiallele in comparison to *clk* induced epialleles to explain the differences in the phenotype. Even between the induced *clk* epialleles that show a spectrum of *superman* to *supersex* phenotypes, epiallele specific methylation patterns cannot be ascertained[41]. It is attributed to mosaic methylation patterns that may vary among the cells of a tissue[41]. In conformity with this study, we do find heterogeneity in the methylation patterns in *lol-1* locus ascertained from our high throughput whole-genome bisulfite genome sequencing data and Chop PCR assay (Fig. 2d). Intriguingly, the induced (*clk*) epialleles show semi-dominance in F1 and revert to WT at a low frequency[25] whereas *lol-1* natural epiallele behaves as a typical mendelian recessive allele, and no revertants were observed among the plants (*n* = ~1200) tested. The *lol-1* epiallele affects only the carpel whorl with normal stamen whorl in contrast to induced *clk* epialleles, affecting both stamen and carpel whorls to varying degrees[6].

**Identification and characterization *lol* epialleles displaying silent to strong *superwoman* phenotypes in geographically distinct natural diploid accessions.** To examine whether hypermethylation of the *SUP* locus is unique to tetraploid Wa-1 accession, we analyzed the methylation landscape of the same locus in 1028 global collections of *A. thaliana* natural accessions

sequenced in 1001 epigenome project[21]. Here, we identified 11 additional diploid accessions showing varying degrees of DNA methylation in *SUP* locus (Fig. 2b, e and Supplementary Fig. 5). Among them, the Geg-14 accession (*lol-2*) displayed phenotypes reminiscent of *superwoman*, *supersex*, and *superman*, all in one inflorescence (100% penetrance; Table 1, Supplementary Fig. 6), with the former two phenotypes being predominant (Fig. 3a–f); in nine accessions, (*lol-3 to lol-11*, Table 1), though a first glimpse gave an impression of WT phenotypes, a closer silique by silique inspection revealed mild *lol* phenotypes (Fig. 3g–k and Supplementary Fig. 6). However, the phenotype is restricted only to few siliques in the inflorescence of some, but not all the plants suggesting low penetrance and expressivity (Table 1). In one accession (Basta-2), consistent with mild methylation, we failed to detect any visible inflorescence phenotypes (Fig. 3l) and thus designated it as a silent *SUP* epiallele (*lol-12*). The hypermethylation state of the *SUP* locus in the natural accessions identified from the 1001 epigenomes project was further corroborated by Chop PCR assay. In five accessions (Wa-1, Geg-14, IP-Trs-0, LIN S-5, and Got-22), there was no PCR amplification of the McrBC digested DNA template in patch-2 consistent with McrBC cleaving these hypermethylated regions. However, in the remaining seven accessions, we observed fainter PCR amplification in the digested DNA template compared to the corresponding undigested controls and digested unmethylated Col-0 control (Fig. 3m and Supplementary Fig. 13b). It suggests that at least this region of *SUP* locus may be heterogeneously methylated in these accessions, consistent with the manifestation of moderate to silent *lol* phenotypes. Similar and complementary results were obtained for the other MSRE, Dde1 (Fig. 3n and Supplementary Fig. 13c). Overall, we find that the more the number of methylated cytosines, the higher the penetrance and vice versa (Fig. 2e and Table 1).

**Suppression of DNA methylation by 5-Azacytidine is not sufficient to restore WT phenotypes in Wa-1.** Genome-wide DNA methylation can be altered by treating *A. thaliana* with the drug 5-Azacytidine[45,46]. To examine whether Wa-1 plants exposed to 5-Azacytidine can restore WT inflorescence phenotypes by inducing loss of methylation at *SUP* locus, we germinated Wa-1 seeds in 5-Azacytidine medium (50, 75 μm) and continued the treatment by fertigation in the soil-grown plants. The treated seeds showed delayed germination, and the plants displayed altered flowering times, plant height, and less biomass compared to control Wa-1 plants concurring with the phenotypes

**Table 1 Phenotypic analysis of natural *SUP* epialleles (*lol*) from diploid natural *A. thaliana* accessions.**

| S. no | Accession name (ABRC Seed Stock No.) | Country of Origin | Number of plants analyzed | No.of plants showing phenotype (% penetrance) | No.of *superwoman* siliques/Total siliques | Phenotypic strength | Allele designation |
|---|---|---|---|---|---|---|---|
| 1 | Geg-14 (CS76876) | Armenia | 25 | 25 (100) | 441/710 | Strong | *lol-2* |
| 2 | IP-Trs-0 (CS77387) | Spain | 28 | 20 (71.4) | 49/879 | Moderate | *lol-3* |
| 3 | Me-0 (CS76549) | Germany | 20 | 9 (45.0) | 28/687 | Moderate | *lol-4* |
| 4 | Cnt-1 (CS78782) | UK | 16 | 5 (31.3) | 9/785 | Weak | *lol-5* |
| 5 | LIN S-5 (CS77040) | USA | 27 | 8 (29.6) | 15/986 | Weak | *lol-6* |
| 6 | Sorbo (CS78917) | Tajikistan | 22 | 4 (18.2) | 14/1140 | Weak | *lol-7* |
| 7 | Tscha-1 (CS76616) | Austria | 26 | 4 (15.4) | 22/1167 | Weak | *lol-8* |
| 8 | Pla-0 (CS76573) | Spain | 19 | 1 (5.3) | 16/1121 | Weak | *lol-9* |
| 9 | IP-Mos-1 (CS77108) | Portugal | 25 | 1 (4.0) | 65/1168 | Weak | *lol-10* |
| 10 | Got-22 (CS76884) | Germany | 26 | 2 (8.0) | 9/729 | Weak | *lol-11* |
| 11 | Basta-2 (CS76692) | Russia | 25 | 0 (0) | 0/720 | Silent | *lol-12* |

of 5-Azacytidine exposed *A. thaliana* plants[47]. Upon flowering, we failed to detect any differences in the floral phenotypes between the 5-Azacytidine treated ($n = 42$) and control Wa-1 plants ($n = 32$)(Fig. 4a, b), further validated by Chop PCR (Fig. 4c and Supplementary Fig. 13d) suggesting that suppression of methylation by 5-Azacytidine is not sufficient to restore WT phenotype. This can be explained by the fact that 5-Azacytidine exerts its demethylating effect by interfering with the function of DNA methyltransferase *MET1*[48]. It is known that *MET1* is not required for maintenance of methylation at *SUP* locus[38], in line with our observation that 5-Azacytidine has no effect on methylation at *SUP* locus.

**Genetic mutations in *CMT3* and *KYP* suppress Wa-*superwoman* phenotypes.** Mutations in the DNA and histone methyltransferases, *CMT3* and *KYP* respectively, are known to suppress *clk* phenotype by causing loss of CHG methylation at *SUP* locus in *clk* epialleles[31,35]. To check whether *lol* phenotypes are suppressed in a similar manner, we generated homozygous double mutants of *lol-1;cmt3-7* ($n = 4$) and *lol-1;kyp-2* ($n = 6$) combinations. All the double mutants displayed WT phenotypes (Fig. 4d, e), indicating that natural *lol* epiallele is subjected to the same epigenetic regulation as induced *clk* epialleles.

**Natural genetic variation in known genes involved in *SUP* locus DNA methylation is not associated with *superwoman* (*lol*) phenotypes.** To ascertain whether any natural genetic variation common to these 12 accessions correlates with methylation of *SUP*, we first screened the 1001 genome database for genetic polymorphisms in known candidate genes *CMT3*, *KYP*, and *AGO4*, that are directly involved in the methylation of *SUP*. Our analysis revealed no significant association linking *SUP* methylation with natural allele variations existing in these three genes (Supplementary Figs. 7, 8). Notably, the G to A SNP seen in the Wa-1 *SUP* locus is not linked with *lol* phenotypes. Despite showing *lol* phenotypes, the rest of the accessions (*lol-2* to *lol-12*) lack this variant as inferred from the 1001 genome project and validated by dCAPS assay (Supplementary Fig. 9).

Next, we examined the possible link of *CMT2*, whose involvement is unknown in *SUP* locus methylation. CMT2-dependent pathway mediates the bulk of DNA methylation in CHH context, especially in the transposon-rich pericentromeric regions[30,49]. Genetic variation at *CMT2* gene locus in natural accessions, especially the truncated $cmt2_{STOP}$ alleles, is shown to be significantly associated with decreased (~1%) genome-wide CHH methylation levels identical to *cmt2* knockout mutants[50,51]. As the penetrance of different *lol* allele phenotypes is proportional to the number of cytosines methylated, and a notable number of

cytosines in the *SUP* locus are in CHH context (Fig. 2b, e), we analyzed for the possible association of the natural variation at *CMT2* locus in all of the 12 accessions harboring *lol* epialleles (Supplementary Fig. 10a, b). Three of the twelve accessions (Geg-14, Sorbo, and Basta-2) contain the $cmt2_{STOP}$ allele (Supplementary Fig. 10a), which may be invoked to explain the reduced DNA methylation and weak penetrance of *lol* phenotypes observed in Sorbo (*lol-7*) and Basta-2 (*lol-12*) accessions. However, Geg-14 (*lol-2*) accession, which also harbors the truncated null allele, shows strong *lol* phenotypes with 100% penetrance. Hence, like other methyltransferase MET1, CMT2 is also not required for maintenance of *SUP* methylation. Besides, other *lol* accessions (for, e.g., Got-22, Ip-Mos-1, and Pla-0) with WT *CMT2* allele also show a weak phenotype. Further, for reasons unknown, all $cmt2_{STOP}$ alleles (depending on the natural accessions) do not behave as null alleles[50], suggesting natural variation at *CMT2* locus in *lol* accessions is unlikely to be a major player in determining the strength of the *lol* epiallele. Finally, phylogenetic analysis using whole-genome SNP polymorphisms revealed that most of the *SUP* hypermethylated accessions are genetically distant and dispersed, suggestive of probable, multiple independent origins of hypermethylated *SUP* locus in diverse niches (Supplementary Fig. 11).

**Stable meiotic transmission and absence of *trans*-chromosomal spread of DNA methylation between *lol* epialleles and WT alleles.** In an F1 hybrid, a methylated allele can trigger *trans* spreading of the methylated state to an unmethylated allele by trans-chromosomal methylation[52]. Using a combination of dCAPS and Chop PCR in the Col-0 × Wa-1 F1 and F2 progeny, we show that the hypermethylated Wa-1 (*lol-1*) epiallele and unmethylated Col-0 *SUP* allele remain unaffected without any reciprocal transmethylation/demethylation effects. Retention of stable epigenetic states in F1 plants is consistent with the recessive behavior of *lol-1* epiallele displaying a loss of *lol-1* phenotypes in those hybrids (Supplementary Fig. 3a). The respective methylation states of the parental alleles were transmitted intact in the F2 progeny (Fig. 4g and Supplementary Fig. 13f), co-segregating with the mendelian phenotypic ratio of 3:1. We also observed similar results in another F2 population derived from a heteroallelic F1 hybrid carrying a strong (*lol-1*) and weak (*lol-7*) epialleles, respectively (Fig. 4h, i and Supplementary Figs. 12, 13g, h). Further, we also show that transgenic Col-0 *SUP* genomic locus that complements the *lol* phenotype in tetraploid Wa-1 plants also remain unmethylated (Fig. 4f and Supplementary Fig. 13e). These results, collectively, rule out *trans*-chromosomal spread of methylation between endogenous *SUP* alleles and ectopic transgenic locus.

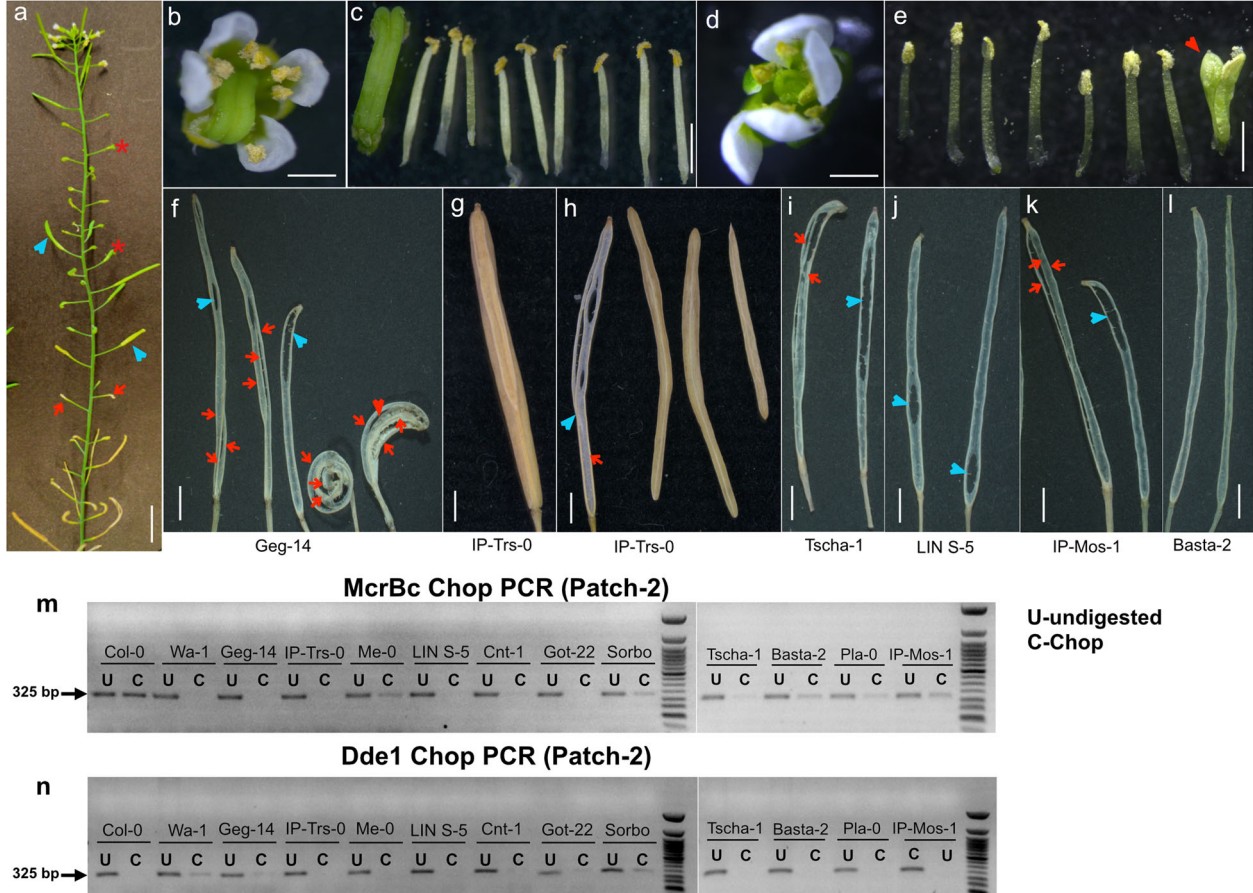

**Fig. 3 The spectrum of phenotypes seen in *SUP* hypermethylated diploid natural accessions. a** A representative Geg-14 accession inflorescence depicting a spectrum of *sup* phenotypes: *superwoman* (blue arrowhead), *superman* (red arrow), and *curly siliques*(red asterisk). **b** A flower from Geg-14 inflorescence with *supersex* phenotype. **c** Dissected third and fourth whorls of the flower **b** respectively showing nine anthers and three carpels. **d** A flower from Geg-14 inflorescence with *superman* phenotype **e** dissected third and fourth whorls of the flower shown in **d**, respectively, showing seven anthers and a pistillode (sterile pistil) as indicated by arrowheads. **f** Replum–septum skeletons of dehisced silique from Geg-14 showing multiple replums (red arrows) and abnormal fusion of septum (blue arrowheads). **g** A partial tricarpellary silique from IP-Trs-1 plant. **h** The silique shown in **g** is split open to reveal its anatomy. The silique is bicarpellary and normal (red arrow) at the start, however in the midway, one of the repla bifurcates (blue arrowhead) to produce a partial tricarpellary silique (three carpel valves are shown). **i–k** The spectrum of replum–septum skeleton phenotypes from various accessions (as indicated) that show weak *superwoman* phenotypes. **l** Replum-septum skeleton from the *SUP* silent Basta-2 accession showing WT phenotypes. **m** Chop PCR assay in the natural accessions with hypermethylated *SUP* locus using the McrBC enzyme. The strong *lol* epialleles from Wa-1, Geg-14, and the moderate *lol* epialleles from IP-Trs-0 and the weak *lol* epiallele from Got-22 are fully methylated at cytosines at McrBC recognition sites, as they are PCR negative for the primers flanking patch-2. The remaining accessions show a faint PCR signal compared to their respective undigested controls suggestive of heterogeneous methylation of cytosines at this site. **n** Chop PCR assay in the natural accessions using the Dde1 enzyme. The strong *lol* epialleles from Wa-1, Geg-14, and the weak *lol* epialleles from Got-22 and Sorbo contain either fully or partially methylated cytosines at Dde1 recognition sites, as they are PCR positive for the primers flanking patch-2. The remaining accessions are PCR negative suggestive of the absence of cytosine methylation at this site. DNA markers used in the gels is 50 bp NEB ladder. Scale bars: **a** = 1 cm, **b–l** = 1 mm.

**SUP mRNA expression is unaltered in the *lol* natural epimutants showing *superwoman* phenotypes.** To test whether hypermethylation of *SUP* locus causes a reduction in *SUP* mRNA levels as reported for *clk*-3 epiallele[25], we carried out real-time qRT-PCR on the Wa-1 (*lol-1*) and Geg-14 (*lol-2*) accessions that showed strong *Wa-superwoman* phenotypes. Surprisingly, we didn't find any changes in mRNA levels both in diploid Wa-1 and Geg-14 compared to unmethylated Col-0 control (Fig. 4j). Our results are in corroboration with an earlier study reporting no reduction in the mRNA levels in some of the induced *sup* epialleles[6]. We also failed to detect novel splice variants unique to *lol* accessions other than the one previously reported common to both WT and induced epimutants[6] (Fig. 4k and Supplementary Fig. 13i, j). In the absence of no difference in total *SUP* mRNA-levels between WT and *lol* epimutants, and *SUP* mRNA being expressed only in a narrow window of time, restricted to third and fourth whorls of the floral meristem[3–8], it would be challenging to detect spatio-temporal quantitative fluctuations of mRNA, if any, by RNA in situ hybridization. This one of our limitations to account for the phenotypes in *lol* accessions despite similar WT mRNA levels. Besides, heterogeneous methylation of *SUP* locus among the cells also is a confounding factor that may or may not affect mRNA levels. Another post-transcriptional possibility to account for the phenotypes is the quantitative fluctuations in SUP protein levels despite normal levels of mRNA[6] or *SUP* locus is partially transcribed, resulting in a truncated mRNA/protein. However, attempts to raise quality antibodies against SUP for immunolocalization studies remain unsuccessful[6], making it challenging to address this hypothesis.

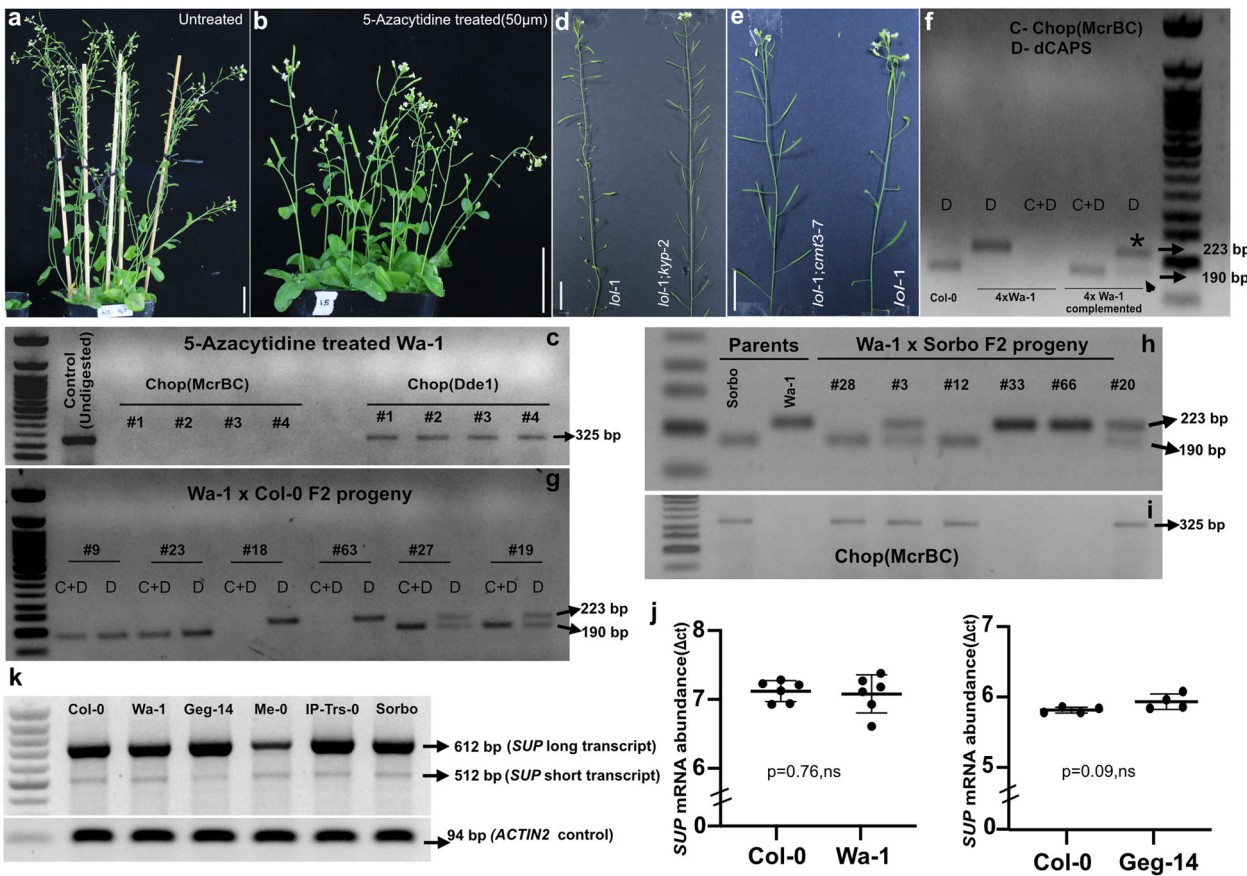

**Fig. 4 SUP mRNA expression and DNA methylation analysis. a, b** 5-Azacytidine treatment of Wa-1 plants does not suppress the silencing of *lol-1* phenotypes. The Wa-1 (*lol-1*) floral phenotypes (from ~50-day old plants) are identical both in the untreated control (**a**) and 50 μm treated 5-Azacytidine plants (**b**). **c** Chop PCR assay with McrBC to confirm the *SUP* locus's unchanged methylated state in 5-Azacytidine-treated plants. **d, e** Mutations in the histone methyltransferase *KYP* and DNA methyltransferase *CMT-3* restore the WT phenotypes in *lol-1* plants. Representative homozygous double mutant combinations of *lol-1;kyp-2* (**d**) and *lol-1;cmt3-7* (**e**) showing WT inflorescence phenotype in comparison to *lol-1* plant. **f–i** *Trans*-chromosome methylation analysis in the F2 segregating population and the complemented tetraploid Wa1 plants. **f** Chop PCR(C) assay reveals the absence of trans chromosomal methylation in complemented tetraploid Wa-1 plants. Control 4x Wa-1 plants are C and dCAPS(D) assay negative, whereas 4x Wa-1 complemented plant with Col-0 *SUP* genomic locus is PCR positive only for Col-0 allele after C + D assay. The D assay alone on 4x Wa-1 complemented plant reveal heterozygosity with strong PCR signal for 4x Wa-1 allele consistent with its abundance (as indicated by asterisks) compared to the weak signal of ectopically inserted Col-0 clone (as indicated by arrowhead). **g** C and D assay in the patch-1 of *SUP* locus from representative F2 progeny from Wa-1 × Col-0 segregating population. Plants(#9,#23) displaying WT phenotype show Col-0 unmethylated allele both in D and C + D assay (190 bp fragment), indicating they are homozygous for Col-0 allele. Plants (#18, #63) displaying the Wa-1 inflorescence phenotype show Wa-1 unmethylated allele both in D and C + D assay (223 bp fragment), indicating they are homozygous for Wa-1 allele. Plants (#19, #27), display WT phenotype, but heterozygous in D (both 223 and 190 bp), and PCR positive only for uncut, unmethylated Col-0 allele after McrBC digestion. **h, i** C and D assay in the patch-2 of *SUP* locus from representative F2 progeny from Wa-1 × Sorbo F2 segregating population. Plants (#12, #28) display WT phenotype confirmed by D assay and C positive with McrBC digested template. Plants (#33,#66) display Wa-1 inflorescence phenotype, confirmed by D assay showing homozygosity for Wa-1 allele and C assay negative. Plants (#3,#20) display WT phenotypes, are heterozygous as revealed by D assay and C positive only for Col-0 allele. **j** A dot plot of $\Delta^{ct}$ values representing *SUP* mRNA abundance, with respect to *ACTIN2* as internal control in hypermethylated Wa-1 and Geg-14 accessions, compared to unmethylated WT Col-0 control (unpaired *t*-test (two-tailed) $p = 0.76$ (Col-0 and Wa-1), $p = 0.09$ (Col-0 and Geg1), ns non-significant, $a = 0.05$). Six and four biological replicates for Wa-1 and Geg-14, respectively, were analyzed. **k** Semi-quantitative RT-PCR showing the abundance *SUP* mRNA and alternative splice variant in the control Col-0, and representative *lol* accessions. *ACTIN2* is used as an internal control. Error bar = Standard deviation (SD) of biological replicates. DNA markers used in the gels is 50 bp NEB ladder. Scale bars: **a, b** = 2 cm; **d, e** = 1 cm.

## Discussion

In this study, we have identified an array of wild *SUP* epialleles (*lol-1* to *lol-12*) with a spectrum of inflorescence phenotypes from 12 natural accessions spread across three different continents. These natural *lol* epialleles constitute a multiple *SUP* (epi)allelic series, along with known induced *clk* epialleles and mutant genetic alleles. As is the case with multiple alleles, we do observe a dominance hierarchy in the epiallelic series. The hierarchy is as follows: WT *SUP*(dominant) > weak *lol* epialleles > strong *lol* epialleles > strong *clk* alleles. In different cross combinations involving any two epialleles, the dominant epiallele expresses its phenotype in F1 and segregates in a mendelian fashion in the F2 generation (Supplementary Figs. 3, 4, 12). These *lol* epialleles behave as loss of function genetic alleles with stable meiotic transmission, at least for three generations studied. However, the *lol* epialleles show variable expressivity in the floral phenotypes in the absence of genetic heterogeneity within an accession. This can be attributed to stochastic, labile epigenetic states leading to mosaicism between the cells or between different individuals causing variable phenotypic expression[53] as observed in mammalian epialleles[54,55].

It is not known what triggered the origin of these natural *lol* epialleles in certain wild accessions, but its stable maintenance and transmission require the action of known molecular players of *SUP* methylation such as *KYP* and *CMT3* methyltransferases. The establishment of de novo DNA methylation is effected by siRNA mediated RNA dependent DNA methylation pathway[56]. However, siRNAs from *SUP* locus have not been detected either in WT or *clk* epimutants[36]. Further, if DNA methylation is established through the siRNA mediated RdDM pathway, then there should be *trans* spreading of methylation from the hypermethylated epiallele to the unmethylated allele in F1 hybrids[52,57] and the F2 segregants[58,59] as well as in transgenic lines with the ectopic insertion of *SUP* locus. However, our results do not conform with this proposition, ruling out the involvement of *SUP* siRNAs in the DNA methylation of *SUP* locus[36]. We failed to find any significant association of natural variation for known genes involved in differential methylation of *SUP* locus, implying likely involvement of hitherto unknown factors controlling methylation of *SUP* locus in these accessions. Hence, the genomes of these natural *lol* epimutant accessions may hold a key towards unraveling the missing links in the epigenetic regulation of *SUP*.

An independent origin and prevalence of such natural epialleles of *SUPERMAN* in a dozen wild diploid and tetraploid populations cannot be ignored as stochastic events, but assumes significance when viewed in terms of the evolution of sex determination systems in plants. Based on the imperfect unisexuality phenotypes (*superman* vs. *superwoman*) manifested by allelic/epiallelic series of *SUP*, it is posited that alteration of the *SUP* or *SUP*-like genes is likely to be involved in the origin of flower bisexuality[6]. Orthologs of *SUP*/*SUP*-like genes from both diecious *Silene latifolia* and monecious *Cucumis sativus* are expressed only in female flowers, implying its involvement in sex determination by differential gene expression[60,61]. A flexible mechanism to achieve tissue-specific differential gene expression without altering the DNA sequence is by epigenetic alteration[62]. Such a mechanism underlying sex differentiation in flowering plants has been postulated[63], and subsequently proven in persimmons and melon[14,62]. Unisexuality has evolved independently, multiple times in flowering plants, with wide variation between species implying different genetic basis and mechanisms[64]. Hence, we speculate that the origin of epialleles in the floral boundary gene, such as *SUP*, may be an early event in the evolutionary path towards unisexuality in specific flowering plant lineages. Notably, the *SUP* locus seems to be a vulnerable hotspot for epigenetic modification by hypermethylation, as revealed by the predominance of both artificially induced and natural epialleles (from this study) isolated for any gene loci in *A. thaliana*.

Interestingly, almost all natural *lol* epialleles (11/12) portray *superwoman* phenotypes. In striking contrast, most of the induced epialleles (10/12), are predominantly *superman* or *supersex*[6], except for *carpel*[41] and *epi3A1*[6]. This suggests that natural epialleles have evolved towards altering later rather than early functions (before vs. after carpel meristem specification respectively) of *SUP*, thus confining the phenotype only to the carpel whorl giving rise to *superwoman* features. This observation calls into question why the preponderance of *superwoman* phenotypes in natural populations? We interpret this in terms of the enhanced reproductive fitness of *superwoman* over *superman* for its existence in the wild. A model for evolutionary transition from hermaphroditism to unisexuality predicts a two-step pathway[65]. The first step is to create females with a reproductive advantage over hermaphrodites, a system known as gynodioecy. In the second step, the males evolve by supplanting the hermaphrodites leading to dioecy. In this aspect, being a *superwoman* (super fertile) has a selective advantage over *superman* (which is sterile), at least in the early stages, to maintain the transmittance of

altered epigenetic states until stable gynodioecy evolves in the lineage.

The epialleles from diploid accessions show mild hypermethylation and less penetrance than the densely hypermethylated tetraploid Wa-1 accession. The evolution of unisexuality is often associated with stable polyploidy in several plant genera[65]. The process of polyploidization, as a consequence, leads to a series of genetic, epigenetic changes, and chromosomal rearrangements[66]. Hence, we hypothesize that polyploidization event might trigger the transition of mild epigenetic states prevailing in the diploids to dense hypermethylated states causing stronger phenotypes in polyploids. The existence of such a metastable *superwoman* phenotype in one of the rarely existing natural tetraploid *A. thaliana* accession (Wa-1) supports this line of thought. It may be noted that our interpretation linking the existence of natural *lol* epialleles of *SUP* in wild populations with the evolution of unisexuality in plants is speculative, inferred based on existing literature, and thus requires further evolutionary studies to validate our conclusions.

In conclusion, we propose that the natural *sup* epialleles (*lol* series) captured here are likely to be evolutionary vestiges or transition intermediates in the early evolutionary path leading to unisexuality, at least, in the *Arabidopsis* lineage. In the family Brassicaceae, *Lepidium sisymbrioides* is the only known diecious polyploid sister genus to *A. thaliana*[67]. *L. sisymbrioides* have diverged relatively recently from *A. thaliana* lineage, and unisexuality (dioecy) has evolved as a result of selective abortion of reproductive whorls at the same floral developmental stage from a bisexual flower[67]. A possible role of *SUP* in the diminution of reproductive whorls of *Lepidium* has been contemplated[68,69]. Hence, it will be interesting to see epigenetic regulation of *SUP*, if any, in the evolution of dioecy in *L. sisymbrioides*.

## Materials and methods

**Plant materials and growth conditions**. Plants were grown in pots under fluorescent lights (7000 lux at 20 cm) at 20 °C with a 16 h light/8 h dark cycle at 70% RH in a controlled growth cabinet (Percival Inc.) or in walk-in growth rooms (Conviron). All the diploid natural accessions (CS 78917(Sorbo), CS78782(Cnt-1), CS77387(IP-Trs-0),CS77108(IP-Mos-1),CS77040(LIN S-5), CS76884(Got-22), CS76876(Geg-14), CS76616(Tscha-1), CS76573(Pla-0), CS76549(Me-0)), tetraploid Wa-1 accessions (CS22644, CS28805, CS39006, CS6885, CS1586) and *superman* mutants: sup-5(CS3882), flo10-1/sup-2(CS6225), clk-3(CS69095), cmt3-7 (CS6365), kyp-2(CS6367) used in the study was obtained from the *Arabidopsis* Biological Resource Center (ABRC), Ohio State University (OSU). WT Col-0 and L*er* stocks used in this study were a gift from Late Dr. Simon Chan, University of California, Davis. The Wa-1 ploidy series consisting of triploid ($2n = 3x = 15$), diploid ($2n = 2x = 10$) and haploid ($2n = x = 5$) plants were generated by CenH3-mediated genome elimination method as described[40,70].

**Microscopy and imaging**. The siliques and flower phenotypes were examined either using Carl Zeiss Stemi 2000-C stereo zoom trinocular microscope or Leica M205 FA stereomicroscope. Images were captured using Zeiss Axiocam 105 or Leica DFC30 FX cameras attached to its respective microscopes. The inflorescence and whole plant images are captured using Canon 70D SLR camera. The images are edited either in Adobe Photoshop CS6 and Affinity Designer.

**Phenotypic analysis of Wa-1 accessions**. Several Wa-1 germplasm stocks (CS28805, CS39006, CS6885, CS1586) deposited by several independent research groups exist at ABRC, OSU. Hence, to ascertain whether the Wa-*superwoman*(*lol-1*) phenotype is limited to CS22644 stock, we grew ~50 plants each from all the four Wa-1 stocks and scored for the Wa-*superwoman* inflorescence phenotypes.

**Genetic mapping of *SUP* locus**. Using the TAIR polymorphism/allele search tool (www.Arabidopsis.org), a set of 53 genome-wide, evenly distributed polymorphic markers (SSR and indel) that can distinguish Wa-1 from Col-0 and L*er* accessions were shortlisted. Upon experimental validation by PCR genotyping, 47 markers polymorphic between Col-0 and Wa-1 (Supplementary Fig. 3b, c), 33 markers between L*er* and Wa-1 (Table S3). For segregation analysis of Wa-*superwoman* phenotypes, Wa-1 derived diploid was crossed to both Col-0 and L*er* accessions. The resultant F1 hybrids were further confirmed by genotyping using a subset of validated polymorphic markers. For genetic mapping, seeds from self-pollinated Wa-1 × Col-0 F1 plants were pooled and advanced to raise an F2 segregating

population. Out of 520 F2 progeny, 126 plants were found to show *lol-1* phenotype. DNA was extracted individually from all the 126 plants for mapping the causative locus. For rough mapping, we employed bulk segregant analysis[71] and identified a marker MSAT3.19 (physical coordinate 8808167) the *q* arm of the third chromosome that is tightly linked with the phenotype. Next, the MSAT 3.19 marker was genotyped individually in 100/126 F2 plants and found 89 plants homozygous for the Wa-1 allele, and the remaining 10 were heterozygous, confirming its proximity to the candidate locus. We chose *SUP* as the candidate locus for the reasons mentioned in the "Results" section.

For fine mapping, the SNP (G to A) in Wa-1 *SUP* locus is exploited to derive a PCR based dCAPS (derived Cleaved Amplified Polymorphic Sequences) marker for distinguishing Wa-1 allele from Col-0/L*er* allele. A set of primers (MR692 Forward: 5′TTTCTTTAAAGTTTCATTTTATTAAATCT**CAT**AT 3′; MR693 Reverse: 5′ AAGATCTGAAAAGATGAACTCAC3′) were designed to amplify a 223 bp fragment encompassing patch-1 of *SUP* locus (Fig. 2a, c) in a way that the forward primer generates a *NdeI* restriction site in combination with the Col-0 SNP but not with Wa-1 SNP. Hence, upon *Nde1* restriction digest of 223 bp amplified product, the Wa-1 allele remains uncut, in contrast, the WT allele gets cut into 190 bp and 33 bp fragments, which can readily be distinguished by agarose (2.5%) gel electrophoresis.

**Plasmid construction, plant transformation, and complementation analysis**. The 6.7 kb *SUPERMAN* genomic region (corresponding to physical coordinates 8237177 to 8243854, TAIR 10), which complements the *sup* genetic as well as epigenetic mutation[2,25], was PCR amplified from both WT Col-0 and Wa-1 accession with polymorphic SNP and cloned into pCAMBIA1300 as a *SacI* and *PstI* fragment using the following primer combinations; Forward: 5′gaac**GAGCTC**agc-taccatattataatacg3′ and reverse: 5′gtaa**CTGCAG**tgcccttgtataggaattc3′. The resultant binary vector was then transformed into *Agrobacterium tumefaciens* strain ASE by electroporation. The transgene construct was introduced into *sup-5* recessive mutant and Wa-1 tetraploid by floral dip method. The seeds obtained from the transformed plants were plated onto a selection medium (1% KNO3 + 0.8% bacto agar + 50 μg/ml hygromycin), and the transformants were selected for hygromycin resistance a week after plating and then transferred to soil for further growth. The transformants were then genotyped to confirm the presence of Wa-1 SNP using the dCAPS marker.

For the allelism test, we crossed Wa-1 derived diploid as a female parent to *sup-5* recessive (CS3882), *flo10-1/ sup-2* (CS6225), and *clk-3* (CS69095) as a pollen parent. The resultant F1 seeds were sown, and after flowering, the *sup* phenotypes were scored in the inflorescence of F1 plants.

**Whole-genome sequencing and bisulfite sequencing analysis**. The genomic DNA from Wa-1 derived diploid plants was isolated using DNA extraction plant kit (ORIGIN) from the tissues pooled from leaves and inflorescence of two independent plants (biological replicates). The genomic DNA was fragmented, end-repaired, and adapters were ligated to the DNA fragments, PCR amplified and sequenced by Illumina's paired-end chemistry using Nextseq 500 at FASTERIS SA (Switzerland) and Eurofins Scientific (India). BBtools-BBduck v38.59 was used for the illumina adapter removal. FASTQC v0.11.7 was used for illumina reads quality check. BWA was used to map the illumina reads to the reference genome from TAIR10. In the case of bisulfite sequencing, adapter-ligated DNA fragments were bisulfite converted using EpiTect Bisulfite Kit (Qiagen),

The de novo assembly of the diploid Wa-1 genome was done using SPAdes assembler. Blast analysis of the resulting contig revealed no structural variations within, and proximal to *SUP* gene except an SNP upstream in the promoter region identical to tetraploid Wa-1 sequence available at 1001 genomes project. The assembled contigs were compared with TAIR10 Col-0 reference for insertions and deletions using Assemblytics and the bed output was then analyzed for any insertion or deletion nearby *SUP* gene. To check for the presence of any unique transposons or transposon-like elements in the Wa-1 accession, we aligned the Col-0 reference genome sequence (TAIR10) with the assembled derived diploid Wa-1 sequence (our sequencing data) and with tetraploid Wa-1 sequence (1001 genomes project data) using Assemblytics1.2. The DNA sequences of the *CMT2*, *CMT3*, *KYP*, and *AGO4* for the 12 accessions are extracted from the 1001 genome project. The sequences were aligned using MEGAX[72] software.

In the case of bisulfite sequencing, DNA sequence 10 kb upstream and 5 kb downstream of the *SUP* gene from TAIR 10 was used as a reference to map the high quality reads using the Bismark package[73]. The raw BS-seq data were filtered using the AfterQC package[74] to obtain high-quality reads. The clean reads were then mapped back to the C to T and G to A converted references using bowtie as the mapping tool. After removing clonal reads, the number of methylated and unmethylated reads along with their trinucleotide context was generated for each cytosine on both the strands. The bisulfite conversion efficiency, as calculated using lambda DNA, was 97.3%. The cytosine report from two biological replicates were merged and methylation call was performed. The cytosines with overall coverage equal or more than five were considered for methylation calling. The cytosine with five or above-methylated reads or more than 40% of the reads were considered as methylated. The data for cytosine methylation of accessions other than Wa-1 diploid is obtained from the 1001 epigenome project[21]. The whole-genome sequencing and bisulfite sequencing data for Wa-1 diploid is submitted to NCBI

under the bio project accession PRJNA633425 consisting of three SRA accession identifiers: SRR12367310, SRR11805321, and SRR11805322.

**Chop PCR analysis**. The DNA from respective genotypes used in Chop PCR assay was extracted using the standard CTAB method. One microgram of DNA (as quantified by Colibri microvolume spectrometer) was digested with methylation-sensitive restriction enzymes McrBC and Dde1 (both from NEB Ltd) at a concentration of 10 units/μg of DNA template in a 30 μl reaction mix, for at least 6 h. After heat inactivation, 4 μl of the digested DNA was used as a template in a 20 μl reaction volume for PCR. Two densely methylated patches (Patch-1 and Patch-2) in *SUP* locus, as indicated in Fig. 2a, c is used for Chop PCR assay. Two MSRE's, namely McrBC (cleaves the site if cytosine is methylated) and Dde1 (protects the site if cytosine is methylated), were chosen for Chop PCR assay. McrBC has at least one recognition site in both Patch-1 and Patch-2, whereas Dde1 has one site only in the patch-2.

For Chop PCR assay involving natural *lol* accessions, we chose patch-2 for analysis to use both complementing MSREs for the assay. The digested and undigested controls are PCR amplified using the primer combinations (MR860:5′CATGGCCACCAGATTACAC3′, MR862: 5′GTCTTACAAGCGTTTTCTGGTGT AT3′) spanning the patch-2 of *SUP* locus to give a 325 bp fragment on PCR amplification.

For *trans* chromosomal methylation experiments involving Wa-1 × Col-0 F2 segregants (Fig. 4g) and Wa-1 tetraploid complemented plants, we employed the dCAPS primer combination (MR692 and MR693) that span the patch-1 as described earlier for mapping experiments to distinguish Wa-1 (*lol-1*) allele from Col-0 *SUP* and other *lol* alleles. This dCAPS assay, when carried out in succession with McrBC digested DNA template helps to distinguish the methylated Wa-1 allele from the unmethylated Col-0 allele (Fig. 4g). However, in the case of Wa-1 × Sorbo F2 segregating population the dCAPS primer combination cannot be employed because both Wa-1 and Sorbo would be cleaved by McrBC, thus cannot distinguish either allele. To overcome this, we used another primer pair MR860 and MR862 (spanning the patch-2 region of *SUP* locus), which in combination with the McrBC digested DNA template, can distinguish the strong Wa-1 (*lol-1*) epiallele from weak (*lol-7*) Sorbo epiallele (Fig. 4h, i).

**Scoring for *superman* phenotypic series**. A flower is classified as *superwoman* if it has typical stamen whorl with six anthers and supernumerary fourth whorl with greater than two carpels; *superman*, if the flower has supernumerary stamens at least greater than seven with sterile, dysfunctional/reduced carpel; *supersex*, if the flower harbors more than six anthers and more than three carpels.

The phenotypes exhibited by natural epialleles are classified as strong if the phenotype shows at least greater than 70% penetrance; moderate, if the penetrance is between 40–70%; weak, if the penetrance is 1–40% and silent if it is less than 1%.

**Double mutant analysis**. Wa-1 diploid plants were crossed with homozygous *cmt3-7* and *kyp-2* as pollen parents. The segregating F2 progeny were genotyped using dCAPS markers for homozygosity of the Wa-1 *SUP* locus. The *cmt3-7* and *kyp-2* mutants were genotyped using the CAPS and dCAPS assay, respectively, using the primers as described[31,35]. The sequences of the primers are given in Table. S4.

**5-Azacytidine treatment**. Surface sterilized diploid Wa-1 seeds were placed in petri dishes containing filter papers soaked in 50 μm, 75 μm concentrations of 5-Azacytidine (Himedia Laboratories LLC), and water (control), cold stratified for 3 days at 4 °C. After cold treatment, the plates were transferred to the growth chamber for germination. Subsequently, the seeds were exposed to the respective concentrations of 5-Azacytidine for every alternate day till transfer to the soil medium. The surviving seedlings were then transferred to the soil medium imbibed with 0.5× MS solution containing the respective concentrations of 5-Azacytidine. After bolting, the inflorescences were sprayed once a week, thrice with respective concentrations of 5-Azacytidine.

**Statistics and reproducibility**. All the statistical analysis described in the study: the chi-square test for goodness of fit for genetic crosses, Kruskal–Wallis (KW) test for seed set data, two-tailed unpaired *t*-test for *SUP* mRNA quantitation was performed using GraphPad PRISM (v 8.4.2) statistical software. We employed the non-parametric KW statistical test, the parametric equivalent to one way ANOVA, for analyzing the seed numbers counted from different types of silique in Wa-1 inflorescence. The non-parametric KW test that is ideal for comparing medians of three or more groups was used here because seed numbers counted from multi-locular siliques didn't fit Gaussian distribution as determined by Shapiro–Wilk test ($\alpha = 0.05$), an essential prerequisite for applying parametric statistical tests. Hence, we chose to summarize the data using the median rather than the mean. Dunn's multiple comparisons test was performed as a posthoc test to analyze statistical significance between different groups of siliques.

**SUP mRNA expression analysis**. Inflorescence heads, a week to 10 days after bolting, were harvested and flash-frozen in liquid nitrogen. Total RNA was isolated

using HiPurA plant RNA extraction kit (HiMedia Laboratories, LLC) as per the manufacturer's instructions. To remove the genomic DNA contamination, the RNA was treated with RQ1 RNase-free DNase (1U, Promega), for 45 min at 37 °C followed by heat inactivation of the enzyme at 75 °C for 15 min. One microgram of total RNA was reverse-transcribed using TAKARA Primescript RT reagent kit as per the manufacturer's instructions. Oligo dT (17mer) primer was used for the first-strand cDNA synthesis. qPCRs were performed on a Mastercycler BIORAD CFX96$^{TM}$ using the primers (MR610 Forward: 5′TTTCTCCTCCATCCTCAC-CAAG3′; MR614 Reverse: 5′TAATCAAACCAATCTCAAGCTC3′) for *SUP* locus and primers (MR372 Forward: 5′TCGGTGGTTCCATTCTTGCTTC3′; MR373 Reverse: 5′TCTGTGAACGATTCCTGGACC3′) for ACT-2 control. The reactions were carried out using TAKARA TB green Premix Ex Taq 2 and incubated at 95 °C for 2 min followed by 40 cycles of 95 °C for 15 s and 60 °C for 30 s. ACT-2 were used for the normalization of all the reactions. PCR specificity was checked by melting curve analysis. At least four independent biological samples were analyzed for each accession as indicated, and qPCR reactions were set up using three technical replicates from each biological replica. *SUP* mRNA abundance was analyzed using normalized Ct ($\Delta^{ct}$) values with ACT-2 as a reference gene.

For semi-quantitative RT-PCR and for checking the splice variants in *SUP* transcripts we used the forward primer MR866 (5′GTCTTACAAGCGTTTTCTGG TGTAT3′) located at the start of 5′UTR of *SUP* gene in combination with reverse primer MR 862 (5′GATATATCTTAGATTTTTCCAGGG3′) flanking the patch-2 of *SUP* locus with same PCR cycling conditions as described for Real-time PCR but only for 27 cycles. ACT-2 was used as an internal control. The resulting PCR products were resolved in a 2% agarose gel.

**Phylogenetic analysis**. The phylogenetic tree was constructed using 50 natural accessions: 12 with hypermethylated *SUP* locus identified in this study and the remaining 38 non-methylated accessions randomly selected from each geographic cluster as classified by 1001 genomes project. The SNP data for all the accessions were downloaded from the 1001 genomes website (https:// 1001genomes.org/data/GMI-MPI/releases/), and they were merged into a single VCF file using BCFtools. Using SNPhylo[75], SNPs were filtered using default values except for the LD threshold which was set at 0.4. The filtered SNPs were collated as a sequence and used MEGA X[72] for phylogenetic tree construction using Maximum likelihood and general time-reversible model (GTR + G) with 100 bootstraps.

**Reporting summary**. Further information on research design is available in the Nature Research Reporting Summary linked to this article.

## Data availability

The whole-genome sequencing and bisulfite sequencing data generated and analyzed during the current study are available from NCBI under the bio project accession PRJNA633425 consisting of three SRA accession identifiers: SRR12367310, SRR11805321, SRR11805322. All other source data are included in the article as supplementary data 1–5.

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

## Acknowledgements

R.B. acknowledges Junior and Senior Research Fellowship for Ph.D. awarded by University Grants Commission, Govt. of India. M.A.B. is the recipient of the INSPIRE Scholarship for Higher Education awarded by Department of Science and Technology (Govt. of India). V.K.P. is a recipient of the Council of Scientific and Industrial Research (CSIR)—Shyama Prasad Mukherjee Ph.D. fellowship. R.M. thanks financial support from Ramalingaswami re-entry fellowship awarded by the Department of Biotechnology (Govt. of India), and Dupont Young Professor grant, Dupont USA. We thank, Indian Institute of Science Education and Research (IISER)- Thiruvananthapuram, for intramural financial and infrastructural support. We thank Puneet Prabhakar Singh for help with the initial characterization of Wa-1 accession, Dilsher Singh Kulaar, for help with Wa-1 × Sorbo crosses, Dr. Ullasa Kodandaramaiah for discussion on phylogenetic analysis, and Dr. Kalika Prasad for sharing the cmt3-7 and kyp-2 seed stocks. We thank Prof. Luca Comai, UC Davis, for providing the facility for growing *A. thaliana* natural accessions and microscopy.

## Author contributions

R.M. conceptualized, designed, and supervised the study. R.B. contributed to certain aspects of the study design and carried out most of the experiments. M.P.M. contributed to the phenotypic analysis of accessions along with R.B. S.S. did the bioinformatics analysis of bisulfite sequencing data along with RB. M.A.B. performed the cloning experiments along with R.B. and partially contributed to the phenotypic screening. V.K.P. contributed to RT-PCR experiments. R.M. wrote the paper with inputs from R.B. M.P.M. helped with editing the draft. All the authors read and approve the draft.

## Competing interests

The authors declare no competing interests.
