## [Peer Review File · Communications Biology]

Reviewers' comments:

Reviewer #1 (Remarks to the Author):

Although increasing evidences demonstrated that spontaneous epimutations may play important roles in plant development and environmental responses on genome-wide scale, there was very few exemplified cases that one single epiallele is responsible for a specific biological function convincingly. By EMS mutagenesis analysis, variable induced epialleles of SUPERMAN was obtained, controlling floral development via hyper DNA methylation of gene body regulated by multiple DNA methylation machineries. However, a natural SUPERMAN epiallele is absent. Here, the authors carried out neat genetic analysis by constructing types of genetic populations and uncovered a natural SUPERMAN epiallele in a tetraploid *Arabidopsis thaliana*. By further screening 1001 epigenome, they also found another 12 natural accessions containing the SUPERMAN epialleles, all of which show superwoman intriguingly. The results are interesting and reliable but will be more convinced if they can optimize the data as following:

1. To make sure the causative effects of methylation level of natural SUPERMAN on flowering phenotype in the natural accessions, they can design methylation sensitive restriction enzymes Chop-PCR analysis, such as MCrBC or MspI/HpaII, for the next genetic analysis.
2. With the above Chop-PCR marker, they can performed the co-segregation analysis of DNA methylation of SUPERMAN with flowering phenotypes using Geg-14XCol-0 F2 population.
3. With the above Chop-PCR marker, they can also analyzed the DNA methylation level of transformed and endogenous loci in complementation experiment to examine the interaction of transgenic locus and endogenous one.
4. Since the high level CHG methylation level of SUPERMAN is controlled by CMT3/KYP4 as previously reported and also observed in the natural one reported , the authors should check the association of natural SUPERMAN methylation variations with the genetic variants of CMT3/KYP4 using 1001 *Arabidopsis* genome/epigenome data.

Minors:

1. Since SUPERMAN is a well-known epiallele made from laboratory with clear epigenetic regulation mechanisms, the authors should mention the background information about plant epigenetics, especially DNA methylation, and highlight the significance of the natural properties of the SUPERMAN in INTRODUCTION and DISCUSSION session.
2. Also, the possible explanations of floral heterogeneity of tetraploid accession should be interpreted in DISCUSSION session.
3. While DNA methylation fail to affect the SUPERMAN mRNA level, the author may think about the possibility that DNA methylation affect the alternative splicing, leading to the flowering phenotypes.
4. In Figure 1k, the authors should label p-values. Also, in Figure 2, the authors should show the exact clone numbers for bisulfite sequencing labeled in figure or mentioned in legend .
5. In Figure S3, I suggest that they may make a mapping graph for rough and fine mapping showing the exact maker position, sample number, physical distance of targeted intervals and candidate gene, which is a popular and clear manner for map-based cloning diagram.

Reviewer #2 (Remarks to the Author):

Bondada et al. addresses the overarching question if epigenetic variation is a driver of phenotypic variation, and if epigenetic variation could play a role in functional evolution. The authors study epiallelic variation at the SUPERMAN locus in *Arabidopsis thaliana*, which is known to determine the formation of sexual organs. Previous studies have shown that chemically induced epiallelic variation at this locus can alter stamen and pistil development, leading to superman, superwoman, or supersex phenotypes.

Starting from a naturally occurring tetraploid *A. thaliana* accession with altered siliques, the authors identified a series of natural accessions with superwoman-like phenotypes. Through a

series of crosses and mapping efforts, the authors narrowed down the genomic region underlying the phenotype to the SUP locus.

Investigation of the SUP locus revealed no significant genetic variants but substantial hypermethylation in these lines, similar to the hypermethylation found in induced epiallelic lines. The authors therefore conclude that these lines carry natural epialleles of the SUP locus, associated with superwoman-like flower morphology.

The paper is interesting in the sense that it describes, for the first time, the existence of natural epialleles of the SUP locus with a phenotypic expression. This nicely complements the genetic and molecular studies that have been published on this locus. The experimental strategy and the data are solid; the findings are presented in a comprehensible way, and the manuscript is overall well written.

Major comments:

- Can the authors exclude the relevance of the SNP that they detected 300 bp upstream of the TSS? It did not become perfectly clear to me whether this SNP is found only in the tetraploid or also in the diploid superwoman accessions.
- How did the authors account for genetic variation? Solely based on the published 1001 genomes data, or did they amplify and sequence the genomic sequence?
- Similar question, relating to line 207 ff: how did the authors verify the absence of a TE?
- Line 198 ff, the authors state that they did not find a methylation pattern that would correlate. I found this statement a bit rushed and would have liked to see a more refined analysis of the pattern correlations between induced and natural epialleles.
- I found the conclusion on SUP epialleles driving the evolution of unisexuality very speculative. It is OK to speculate, but it should be marked more clearly as such.

Minor comments

- The bisulfite sequencing data needs to be made publicly available via a short read archive, in accordance with the journal's requirements. I could not find an SRA accession identifier in the manuscript.
- In the corresponding figures, arrows and arrowheads are hard to discern. It would be easier if the authors used two different colors.
- Line 222 ff: Revise this sentence for syntax.
- I am no expert on flower development terminology, but it seems to me that "fatty" might not be an accurate term to describe this type of silique.
- Line 212: there seems to be a word missing.
- Line 259: this sentence does not seem to make sense.

Reviewer #3 (Remarks to the Author):

In this paper, the authors document the occurrence of natural epialleles at the superman locus. These epialleles modify floral architecture in different ecotypes. This is a cool result of general interest and should be published. However, I have the following suggestions which I think will strengthen several weak points in the narrative of this paper.

Major comments

1. Epialleles must be meiotically stable. Can highly penetrant superwoman phenotypes be recovered in a subset of progeny obtained by selfing the F1 progeny of strong phenotype lines crossed into weak phenotype lines? This experiment would not only strongly establish the epiallelic status but also test if the DNA methylation can spread in trans and silence the active SUP locus.
2. The authors suggest that gene body DNA methylation over the Superman locus is the potential

cause of the epiallele that leads to the phenotype. The authors also find high non-CG methylation at the SUP locus in strains with a strong superwoman phenotype. This high non-CG methylation and middling levels of CG methylation may be linked to differences in RdDM activity. Have the authors checked to see if 21-24nt sRNAs are abundant at the SUP locus in strains with strong phenotypes? Does introgression of an nrpd1, drm2 or CMT2/3 mutation or the removal of DNA methylation via the use of a chemical DNA methylation inhibitor such as azacytidine repress the superwoman phenotype?

3. The authors suggest that the epiallele is independent of RNA expression level. However, this conclusion is not strongly supported. SUP is expressed, as the authors themselves state, in a narrow spatio-temporal band. However, RT-PCR is carried out on whole inflorescences and may thus not provide a clear answer as to whether the gene is transcriptionally affected. The question of whether SUP is transcriptionally silenced is probably better answered by an RNA insitu. Another possibility not detected by RT-PCR is that the SUP locus is partially transcribed in hypermethylated lines and does not produce a full length protein.

The authors should test the RNA levels if possible. However, if they should fail to do so in light of the Covid-19 pandemic, I suggest that the authors note the shortcoming of their conclusion in the paper.

Point by point response to Reviewer's comments

Reviewer #1

Although increasing evidences demonstrated that spontaneous epimutations may play important roles in plant development and environmental responses on genome-wide scale, there was very few exemplified cases that one single epiallele is responsible for a specific biological function convincingly. By EMS mutagenesis analysis, variable induced epialleles of SUPERMAN was obtained, controlling floral development via hyper DNA methylation of gene body regulated by multiple DNA methylation machineries. However, a natural SUPERMAN epiallele is absent. Here, the authors carried out neat genetic analysis by constructing types of genetic populations and uncovered a natural SUPERMAN epiallele in a tetraploid *Arabidopsis thaliana*. By further screening 1001 epigenome, they also found another 12 natural accessions containing the SUPERMAN epialleles , all of which show superwoman intriguingly. The results are interesting and reliable but will be more convinced if they can optimize the data as following:

Response: We thank the reviewer for encouraging comments on our work.

1. To make sure the causative effects of methylation level of natural SUPERMAN on flowering phenotype in the natural accessions, they can design methylation sensitive restriction enzymes Chop-PCR analysis, such as MCrBC or MspI/HpaII, for the next genetic analysis.

Response: We thank the reviewer for suggesting the use of Chop PCR for downstream analysis in genetic crosses involving SUP epialleles. We have now designed and carried out Chop PCR assay using two complementary methylation sensitive restriction enzymes: MCrBC (cleaves the methylated site) and DdeI(protects the methylation site) that is present in SUP locus. We have now employed this assay in the Wa-1 accession to validate it first and then in all the remaining 11 accessions. This is explained in the revised manuscript in the lines 228- 240 and and lines 277-286 and shown in Fig. 2d, Fig. 3m,n

2. With the above Chop-PCR marker, they can performed the co-segregation analysis of DNA methylation of SUPERMAN with flowering phenotypes using Geg-14XCol-0 F2 population.

Response: This suggestion by the reviewer was helpful to strengthen our observation that Wa-1 allele behaves as a mendelian recessive allele, which we previously concluded based on the phenotypic segregation ratio of 3:1 . We now show that the methylation state of the epiallele perfectly co-segregates with the mendelian phenotypes. This is demonstrated by the combined use of Chops PCR and dCAPs assay on two different F2 segregating populations: Wa-1 x Col-0 and Wa-1(strong epiallele) x Sorbo(weak epiallele). This is elaborated in the manuscript in the lines 339- 347. Also please refer to Fig.4g-i and the text for further details.

3. With the above Chop-PCR marker, they can also analyzed the DNA methylation level of transformed and endogenous loci in complementation experiment to examine the interaction of transgenic locus and endogenous one.

Response: In the complementation experiments involving *Wa-1(lol-1)* plants with ectopically inserted *Col-0* and *Wa-1* locus, we do find full genetic complementation as visualized by the restoration of WT phenotypes, indirectly suggesting that the transformed loci may not be vulnerable to *trans* chromosomal methylation. Now we demonstrate the same at molecular level using allele specific dCAPs assay that distinguishes the methylated *Wa-1* allele from the unmethylated *Col-0* allele in combination with Chop PCR. This is explained in the Fig.4f and its legends and in the revised text (Lines 347 -350).

4. Since the high level CHG methylation level of *SUPERMAN* is controlled by *CMT3/KYP4* as previously reported and also observed in the natural one reported , the authors should check the association of natural *SUPERMAN* methylation variations with the genetic variants of *CMT3/KYP4* using 1001 Arabidopsis genome/epigenome data.

Response: We have checked for the association of genetic variants in *CMT3*, *KYP* and *AGO4* genes that are directly linked with methylation of *SUP* loci, using the information available in 1001 genomes data and failed to find any association of the same in all the 12 natural accessions with hypermethylated *SUP*. A detailed analysis of the same is provided in Supplementary information Fig. S7- S9. Further it is explained in the revised text in the lines 316-334

Minors:

1. Since *SUPERMAN* is a well-known epiallele made from laboratory with clear epigenetic regulation mechanisms, the authors should mention the background information about plant epigenetics, especially DNA methylation, and highlight the significance of the natural properties of the *SUPERMAN* in INTRODUCTION and DISCUSSION session.

Response: Thanks for this suggestion. Due to space limitations in the previous version of the manuscript we were unable to elaborate on this aspect. Now we have included a paragraph describing the plant epigenetics, and known facts about the epigenetic regulation of *SUPERMAN* in the introduction (lines 98-132) and in other places in the manuscript as applicable.

2. Also, the possible explanations of floral heterogeneity of tetraploid accession should be interpreted in DISCUSSION session.

Response: We have included new data on the floral heterogeneity observed in tetraploid accession(Lines 154-158). A possible explanation for variable expressivity of the floral phenotypes has been included in the discussion section lines 380-382 as follows.

“the *lol* epialleles show variable expressivity in the floral phenotypes in the absence of genetic heterogeneity within an accession. This can be attributed to stochastic, labile epigenetic states leading to mosaicism between the cells or between different individuals causing variable phenotypic expression⁵³ as observed in mammalian epialleles^{54,55”}

3. While DNA methylation fail to affect the SUPERMAN mRNA level, the author may think about the possibility that DNA methylation affect the alternative splicing, leading to the flowering phenotypes.

Response: We agree there exists a possibility of alternate splicing which may lead to floral phenotypes observed. We checked for alternate splicing in *SUP* locus but didn't detect any novel splice variants other than the one previously reported by which is common to both WT and induced epialleles. We have included this as a separate fig.4k.

4. In Figure 1k, the authors should label p-values. Also, in Figure 2, the authors should show the exact clone numbers for bisulfite sequencing labeled in figure or mentioned in legend .

Response: For Figure 1K: we have done Kruskal Wallis test which is non-parametric equivalent of ANOVA. As the *p* value was significant, we have carried out Dunn's multiple comparison test as the post hoc test and the resulting pairwise *p* values are now indicated in the fig.1k.

We did high throughput whole genome bisulfite sequencing of 2 independent Wa-1 plants described in the figure.2 and the same has been indicated in the line 9 of Figure legend 2 as follows: 2x Wa-1(our high throughput sequencing data, 2 biological replicates). The data for the same has been submitted to NCBI under the bioproject accession PRJNA633425 consisting of three SRA accession identifiers: SRR12367310, SRR11805321, SRR11805322. The reviewer link for the same is as follows:

<https://dataview.ncbi.nlm.nih.gov/object/PRJNA633425?reviewer=9g0kedchhbk9glj3hc7k3re0d>

5. In Figure S3, I suggest that they may make a mapping graph for rough and fine mapping showing the exact maker position, sample number, physical distance of targeted intervals and candidate gene, which is a popular and clear manner for map-based cloning diagram.

Response: Thanks for the suggestion. We have now included the detailed figure as suggested in the supplementary information. Please refer Fig.S3c

Reviewer #2

Bondada et al. addresses the overarching question if epigenetic variation is a driver of phenotypic variation, and if epigenetic variation could play a role in functional evolution. The authors study epiallelic variation at the SUPERMAN locus in *Arabidopsis thaliana*, which is known to determine the formation of sexual organs. Previous studies have shown that

chemically induced epiallelic variation at this locus can alter stamen and pistil development, leading to superman, superwoman, or supersex phenotypes.

Starting from a naturally occurring tetraploid *A. thaliana* accession with altered siliques, the authors identified a series of natural accessions with superwoman-like phenotypes. Through a series of crosses and mapping efforts, the authors narrowed down the genomic region underlying the phenotype to the SUP locus.

Investigation of the SUP locus revealed no significant genetic variants but substantial hypermethylation in these lines, similar to the hypermethylation found in induced epiallelic lines. The authors therefore conclude that these lines carry natural epialleles of the SUP locus, associated with superwoman-like flower morphology.

The paper is interesting in the sense that it describes, for the first time, the existence of natural epialleles of the SUP locus with a phenotypic expression. This nicely complements the genetic and molecular studies that have been published on this locus. The experimental strategy and the data are solid; the findings are presented in a comprehensible way, and the manuscript is overall well written.

Response: We thank the reviewer for the appreciation of our work and for motivating comments.

Major comments:

1. Can the authors exclude the relevance of the SNP that they detected 300 bp upstream of the TSS? It did not become perfectly clear to me whether this SNP is found only in the tetraploid or also in the diploid superwoman accessions.

Response: Yes. We exclude the relevance of the G to A SNP 300 bp upstream by the genetic complementation experiments involving Wa-1 and Col-0 SUP genomic clones, both of which rescue the mutant phenotypes in Wa-1 plants irrespective of the SNP. This is explained in the lines 220-222. Further the G to A SNP is not found in the Geg-14 accession that show strong *lol* phenotypes and also in the remaining 10 accessions that show moderate to weak phenotypes. A detailed analysis of the same is provided in supplementary information Fig. S9

2. How did the authors account for genetic variation? Solely based on the published 1001 genomes data, or did they amplify and sequence the genomic sequence?

Response: The genetic variation for all accessions except Wa-1 was solely based on the published 1001 genome project database, which we further validated using Chop PCR assay.(Fig. 4m,n)

3. Similar question, relating to line 207 ff: how did the authors verify the absence of a TE?

Response: To check for the presence of any unique transposons or transposon like elements in Wa-1 accession, we aligned the Col-0 reference genome sequence(TAIR10) with diploid Wa-1 sequence(in house generated data) and with tetraploid Wa-1 sequence(1001 genomes project data) using Assemblytics1.2. This has been stated in

the materials and methods section (Lines 569-573). Except for a few SNPs there is no evidence of any unique transposons as the alignment is similar to Col-0 reference.

4. Line 198 ff, the authors state that they did not find a methylation pattern that would correlate. I found this statement a bit rushed and would have liked to see a more refined analysis of the pattern correlations between induced and natural epialleles.

Response: Thanks for this clarification. We have now included a supplementary table 3 which indicates the percentage of methylated cytosines shared between Wa-1 and clk epialleles. We have elaborated our analysis in the following sentences in the revised manuscript. (Lines 249 to 258)

“ A majority of the methylated cytosines are common to *clk* and *lol-1* epialleles (~80%, Table S3) except for a few especially in the patch 2 region and others being scattered throughout the loci (Fig.2b,S5). Other than this, we didn't find any characteristic methylation patterns unique to *lol-1* epiallele in comparison to *clk* induced epialleles to explain the differences in the phenotype. Even between the induced *clk* epialleles that show a spectrum of *superman* to *supersex* phenotypes, epiallele specific methylation patterns cannot be clearly ascertained⁴¹. It is attributed to mosaic methylation patterns that may vary among the cells of a tissue⁴¹. In conformity with this study, we do find heterogeneity in the methylation patterns in *lol-1* locus ascertained from our high throughput whole genome bisulfite genome sequencing data and Chop PCR assay (Fig. 2d).”.

5. I found the conclusion on SUP epialleles driving the evolution of unisexuality very speculative. It is OK to speculate, but it should be marked more clearly as such.

Response: We agree with the reviewer that our conclusion linking natural SUP alleles with evolution of unisexuality is speculative. We have now emphasized this as appropriate at various places in the discussion session. In addition, to emphasize it further we have included the following sentences in the conclusion (Lines 440-443).

“It may be noted that our interpretation linking the existence of natural *lol* epialleles of *SUP* in wild populations with the evolution of unisexuality in plants are speculative, inferred based on existing literature and thus requires further evolutionary studies to validate our conclusions”.

Further we have shortened the title of the paper as “ Natural epialleles of Arabidopsis *SUPERMAN* are *superwoman*” by removing the words “its evolutionary implications” from the previous title in support of this comment.

Minor comments

1. The bisulfite sequencing data needs to be made publicly available via a short read archive, in accordance with the journal's requirements. I could not find an SRA accession identifier in the manuscript.

Response: We thank the reviewer for pointing this out. The metadata information that we generated is submitted to NCBI under the bioproject accession PRJNA633425 consisting of three SRA accession identifiers: SRR12367310, SRR11805321, SRR11805322

2. In the corresponding figures, arrows and arrowheads are hard to discern. It would be easier if the authors used two different colors.

Response: Thanks for pointing this out. We have now used two different colors in the figures for easy identification.

3. Line 222 ff: Revise this sentence for syntax.

Response: Thanks for pointing this out. In the revised manuscript this corresponds to lines 275-277. We have revised the syntax “In one accession(Basta-2), consistent with mild methylation, we failed to detect any visible inflorescence phenotypes(Fig.3I) and thus designated it as a silent *SUP* epiallele(*lol-12*)”. We hope this sentence syntax reads fine.

4. I am no expert on flower development terminology, but it seems to me that “fatty” might not be an accurate term to describe this type of silique.

Response: This is a great suggestion. We have now replaced “fatty” with “multilocular” siliques which is in line with the accepted terminology used for describing the reported silique phenotype.

5. Line 212: there seems to be a word missing.

Response: We have added the word “locus”. “we analyzed the methylation landscape of the same locus in 1028 global collections of”. In the revised manuscript the same corresponds to line 266.

6. Line 259: this sentence does not seem to make sense.

Response: We have rephrased the sentence as follows: “Hence, we speculate that the origin of epialleles in the floral boundary gene such as *SUP* may be an early event in the evolutionary path towards unisexuality in certain flowering plant lineages.” In the revised manuscript the same line corresponds to line 410-412.

Reviewer #3

In this paper, the authors document the occurrence of natural epialleles at the superman locus. These epialleles modify floral architecture in different ecotypes. This is a cool result of general

interest and should be published. However, I have the following suggestions which I think will strengthen several weak points in the narrative of this paper.

Response: We thank the reviewer for encouraging comments and support.

Major comments

1. Epialleles must be meiotically stable. Can highly penetrant superwoman phenotypes be recovered in a subset of progeny obtained by selfing the F1 progeny of strong phenotype lines crossed into weak phenotype lines? This experiment would not only strongly establish the epiallelic status but also test if the DNA methylation can spread *in trans* and silence the active SUP locus.

Response: This is an interesting experimental suggestion to check for *trans* chromosomal methylation effects. We now show that *lol* natural epialleles are meiotically stable and behave as recessive genetic mutant alleles. We have analyzed the F1 and F2 progeny of Wa-1(strong epiallele) x Sorbo(weak epiallele) heteroallelic combination and show that the hybrid is displaying Sorbo phenotype indicating that the weak epiallele is dominant over strong epiallele suppressing the strong Wa-1 phenotype (Fig.S12) . Using a combination of Chop PCR and dCAPs assay in the F2 segregants, we show that DNA methylation cannot spread *in trans* to silence the active SUP locus. In addition, we demonstrate the lack of *trans* chromosomal methylation in Col-0 X Wa-1 segregating population and in 4x Wa-1 tetraploid complemented with Col-0 genome clone. A separate figure.4f-i and a paragraph (lines 338-350) describing this results is included in the results as follows:

“In a F1 hybrid, a methylated allele can trigger *trans* spreading of the methylated state to an unmethylated allele by trans-chromosomal methylation⁴⁵. Using a combination of dCAPs and Chop PCR in the Col-0 x Wa-1 F1 hybrids, we found that the hypermethylated *lol-1* epiallele and unmethylated Col-0 SUP allele remain unaffected without any reciprocal trans methylation/demethylation effects. Retention of stable epigenetic states in F1 plants is consistent with the recessive behaviour of *lol-1* epiallele displaying a loss of *lol-1* phenotypes in those hybrids(Fig.S3a). The respective methylation states of the parental alleles were transmitted intact in the F2 progeny, co-segregating with the mendelian phenotypic ratio of 3:1(Fig.4g). Similar results were observed in a F2 population derived from a heteroallelic F1 hybrid carrying a strong(*lol-1*) and weak(*lol-7*) epialleles respectively(Fig.4h,i). Further, we also show that transgenic Col-0 SUP genomic locus that complements the *lol* phenotype in tetraploid Wa-1 plants also remain unmethylated(Fig.4e). Collectively, these results rule out *trans*-chromosomal methylation effects on endogenous SUP alleles and ectopic transgenic locus.”

2. The authors suggest that gene body DNA methylation over the Superman locus is the potential cause of the epiallele that leads to the phenotype. The authors also find high non-CG methylation at the SUP locus in strains with a strong superwoman phenotype. This high non-CG methylation and middling levels of CG methylation may be linked to differences in RdDM activity.

- a. Have the authors checked to see if 21-24nt sRNAs are abundant at the SUP locus in strains with strong phenotypes?

Response: Thanks for this suggestion to examine the epigenetic regulation of natural *lol* epialleles of *SUP*.

We haven't experimentally checked for the presence of 21-24 nt siRNAs as it is known from previous literature that siRNAs are not produced in WT as well as in induced *clk* epialleles of *superman* (Zilberman *et al.*, 2004) that show stronger phenotype than the natural *lol* epialleles discovered here. Further, there is no spread of methylation from strong to weak allele suggesting the absence of *trans* spreading of methylation which is mediated by siRNA mediated RdDM pathway. This we have discussed in the lines 385-397.

- b. Does introgression of an *nprpd1*, *drm2* or *CMT2/3* mutation or the removal of DNA methylation via the use of a chemical DNA methylation inhibitor such as azacytidine repress the superwoman phenotype?

Response: We show that the natural *lol* epialleles follows the same epigenetic regulation known for *SUP* loci. The prevailing model (stated in the lines 127-132) indicates that AGO4 guides KYP to *SUP* chromatin to methylate H3K9, which creates a binding platform for LHP1, which in turn recruits CMT3 catalyzing the methylation of cytosines at *SUP* loci, converting the *SUP* WT allele to *clk* epiallele.

Hence, we generated *Wa-1(lol-1) cmt3-7* and *lol-1 kyp-2* double mutant combinations (Fig. 4d,e) and show that both mutant combinations restores the WT phenotype indicating the functional requirement of both for maintaining methylation at *SUP* locus. This is described in the text (Lines 302-307)

Further, we also demonstrate that 5-Azacytidine treatment is not sufficient to repress silencing of *SUP* (Fig. 4a,b), indicating that MET1 is also not required for maintenance of methylation. (Lines 288-301).

3. The authors suggest that the epiallele is independent of RNA expression level. However, this conclusion is not strongly supported. *SUP* is expressed, as the authors themselves state, in a narrow spatio-temporal band. However, RT-PCR is carried out on whole inflorescences and may thus not provide a clear answer as to whether the gene is transcriptionally affected. The question of whether *SUP* is transcriptionally silenced is probably better answered by an RNA *in situ*. The authors should test the RNA levels if possible. However, if they should fail to do so in light of the Covid-19 pandemic, I suggest that the authors note the shortcoming of their conclusion in the paper.

Response: We agree with the reviewer that RNA *in situ* would be an ideal experiment to examine the transcript levels in the tissues. However, in the absence of no difference in total *SUP* mRNA levels between WT and *lol* epimutants, and *SUP* mRNA being expressed only in a narrow window of time, restricted to third and fourth whorls of the floral meristem, it would be challenging to detect spatio-temporal quantitative fluctuations of mRNA, if any, by RNA *in situ* hybridization. In addition, heterogenous methylation of

SUP loci among the cells also be a confounding factor which may or may not affect mRNA levels. (Lines 366-370).

As suggested by the reviewer we have indicated this as one of the shortcomings in our paper “ This is one of the limitations in our study to account for the phenotypes in *lol* accessions despite identical WT mRNA levels”.(Lines 358-364)

4. Another possibility not detected by RT-PCR is that the SUP locus is partially transcribed in hypermethylated lines and does not produce a full length protein.

Response: Thanks for this suggestion. This is an interesting possibility. We have included this in the manuscript in the lines 366- 367.

REVIEWERS' COMMENTS:

Reviewer #1 (Remarks to the Author):

The revised manuscript is clear and publishable.

Reviewer #2 (Remarks to the Author):

The authors have addressed the criticisms and comments I had raised during the initial review. Most importantly, the authors have clarified some ambiguities regarding the genetic differences at the SUP locus and have attenuated their claims regarding the evolutionary implications of their findings. I have no further comments and think that the manuscript can be published in its revised form.